# Bi-Anchor Interpolation Solver for Accelerating Generative Modeling

Hongxu Chen [1][*]   Hongxiang Li [1][*]   Zhen Wang [1]   Long Chen [1]

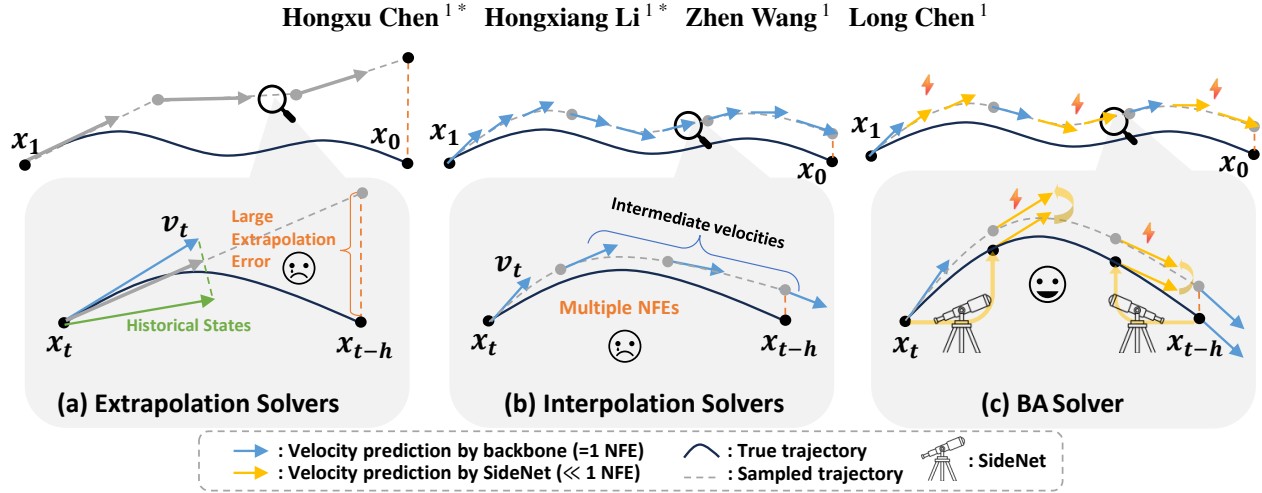

Figure 1. **(a). Extrapolation Solvers** only need a single NFE to calculate current velocity within each interval, but it has large extrapolation error. **(b). Interpolation Solvers** are high-accuracy methods, but they need multiple sequential NFEs within each interval to calculate current and intermediate velocities, which is not efficient for sampling. **(c). BA-solver** utilizes lightweight SideNet to predict intermediate velocities based on bidirectional anchors ($v_t$ and $v_{t-h}$), efficiently providing a high-accuracy approximation for $x_{t-h}$.

## Abstract

Flow Matching (FM) models have emerged as a leading paradigm for high-fidelity synthesis. However, their reliance on iterative Ordinary Differential Equation (ODE) solving creates a significant latency bottleneck. Existing solutions face a dichotomy: training-free solvers suffer from significant performance degradation at low Neural Function Evaluations (NFEs), while training-based one- or few-steps generation methods incur prohibitive training costs and lack plug-and-play versatility. To bridge this gap, we propose the **Bi-Anchor Interpolation Solver (BA-solver)**. BA-solver retains the versatility of standard training-free solvers while achieving significant acceleration by introducing a lightweight SideNet (1-2% backbone size) alongside the frozen backbone. Specifically, our method is founded on two synergistic components: **1) Bidirectional Temporal Perception**, where the SideNet learns to approximate both future and historical velocities without retraining the heavy backbone; and **2) Bi-Anchor**

**Velocity Integration**, which utilizes the SideNet with two anchor velocities to efficiently approximate intermediate velocities for batched high-order integration. By utilizing the backbone to establish high-precision "anchors" and the SideNet to densify the trajectory, BA-solver enables large interval sizes with minimized error. Empirical results on ImageNet-256[2] demonstrate that BA-solver achieves generation quality comparable to 100+ NFEs Euler solver in just 10 NFEs and maintains high fidelity in as few as 5 NFEs, incurring negligible training costs. Furthermore, BA-solver ensures seamless integration with existing generative pipelines, facilitating downstream tasks such as image editing.

## 1. Introduction

Deep generative models have fundamentally revolutionized content creation, with Diffusion Probabilistic Models (DPMs) (Sohl-Dickstein et al., 2015; Ho & Salimans, 2022; Song et al., 2020b) setting the benchmarks for high-fidelity synthesis. More recently, Flow Matching (FM) (Liu et al., 2022; Lipman et al., 2022) has emerged as a unified paradigm. By regressing a velocity field that transports a noise distribution to a data distribution along continuous paths, FM models like Flux (Labs, 2024) and SD3 (Esser et al., 2024) have achieved state-of-the-art performance.

---

[1]The Hong Kong University of Science and Technology. Correspondence to: Long Chen <longchen@ust.hk>.

*Proceedings of the 43rd International Conference on Machine Learning*, Seoul, South Korea. PMLR 306, 2026. Copyright 2026 by the author(s).

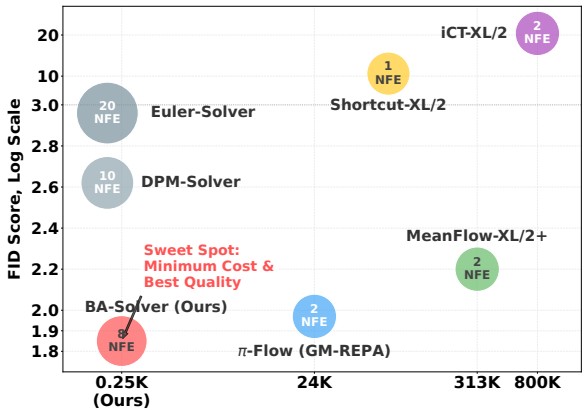

*Figure 2.* **FID, training iteration, and NFEs across methods on ImageNet-**$256^2$**.** BA-solver is located at the sweet spot.

Formally, the generation process in FM models involves solving an Ordinary Differential Equation (ODE), which requires discretizing the continuous time[1] into intervals for numerical integration. For each interval from $t$ to $t - h$, the solver approximates the integral to obtain $x_{t-h}$ given $x_t$. Existing approaches can be categorized into two groups: 1) *Extrapolation solvers* (Lu et al., 2025; Xie et al., 2024): Extrapolation solvers like Euler solver utilize the current velocity ($v_t$), optionally incorporating historical states (*e.g.*, $v_{t+h}$, $x_{t+h}$). However, they suffer from large extrapolation errors facing large interval sizes (*cf.*, Figure 1(a)). To minimize errors, they typically require a large number of *sequential* Neural Function Evaluations (NFEs), creating a significant latency bottleneck for high-fidelity sampling (*e.g.*, 100+ NFEs for Euler solver). 2) *Interpolation solvers* (Zhao et al., 2023; Lu et al., 2022; Zhang & Chen, 2022): Interpolation solvers, including Heun solver, incorporate both the current and intermediate velocities within the interval (*cf.*, Figure 1(b)), achieving higher accuracy and enabling larger interval sizes. Nevertheless, despite the reduced number of intervals, they require sequential multiple NFEs to calculate the current and intermediate velocities in each interval. Consequently, the total NFE count is not efficiently reduced compared to extrapolation solvers (*e.g.*, 20 NFEs for Heun).

In contrast to plug-and-play training-free solvers, recent training-based one- or few-step generation methods, such as Consistency Models (Song et al., 2023; Song & Dhariwal, 2023) and Flow Map Matching (Frans et al., 2024; Geng et al., 2025a; Boffi et al., 2025), have achieved further generation acceleration. Generally, instead of velocity prediction, these methods focus on directly predicting the generation trajectory. This allows them to circumvent iterative ODE solving, thereby condensing inference to just one or a few NFEs. However, they lack the versatility to be directly applied to off-the-shelf models. Additionally, as shown in

---

[1]Throughout this paper, time decreases from $t = 1$ (noise) to $t = 0$ (data).

Figure 2, they incur prohibitive training costs, as shifting the prediction paradigm necessitates extensive retraining or fine-tuning of the parameter-heavy FM model (*i.e.*, backbone), which is neither computationally nor memory efficient.

To bridge the gap between training-free solvers and these training-based one- or few-step generation methods for FM models, we aim to retain the plug-and-play versatility of standard solvers while achieving significant acceleration through lightweight training. To this end, we propose **Bi-Anchor Interpolation Solver (BA-solver)**. BA-solver keeps the backbone frozen, enabling a plug-and-play integration. It introduces a lightweight SideNet (Zhang et al., 2020; Sung et al., 2022) (approximately 1-2% the size of the backbone) that endows the frozen backbone with bidirectional temporal perception. Utilizing *Bi-Anchor Velocity Interpolation*, we enable batched high-order integration within SideNet, achieving a superior balance between efficient and high-fidelity sampling. Specifically, our method is founded on two synergistic components:

**(1) Bidirectional Temporal Perception**: By adhering to the velocity-prediction paradigm, we avoid the need to retrain the backbone. As shown in Figure 1(c), we employ a lightweight SideNet to approximate future and historical velocities conditioned solely on the current states ($x_t$, $v_t$), *i.e.*, *SideNet predictions anchored on* $v_t$. This mechanism provides the backbone with lookahead and lookback capabilities. Since the gradients are prevented from backpropagating through the frozen backbone, the training process is computationally- and memory-efficient. As illustrated in Figure 2, BA-solver requires merely 0.03%–1.0% of training iterations needed by existing training-based methods.

**(2) Bi-Anchor Velocity Integration**: Leveraging SideNet with bidirectional temporal perception, we can achieve high-accuracy interpolation with minimal NFEs. BA-solver utilizes the frozen backbone to establish anchor velocities (bi-anchor) $v_t$ and $v_{t-h}$ at both the start and terminal of each interval. Anchored on $v_t$ and $v_{t-h}$, SideNet can efficiently offer an accurate approximation of intermediate velocities in a batch within each interval (*cf.*, Figure 1(c)). Then, we can densely approximate the fine-grained integral utilizing these velocities, facilitating high-order numerical integration at negligible computational cost. Furthermore, by reusing $v_{t-h}$ in the current interval for the next, BA-solver achieves further reduction of NFEs.

To validate its effectiveness, we conducted comprehensive evaluations across multiple resolutions ($256^2$ and $512^2$). On ImageNet-$256^2$, empirical results demonstrate that BA-solver achieves a remarkable FID of 1.72 with just 10 NFEs, matching the quality of 100+ NFEs Euler solver. It further enables high-fidelity synthesis with as few as 5 NFEs. Crucially, BA-solver necessitates significantly fewer training iterations compared to learning-based methods. Addition-

ally, we demonstrate the effectiveness of BA-solver in image editing through qualitative results.

In summary, our contributions are threefold: **1) Algorithm:** We propose BA-solver, a novel sampling method that enables high-fidelity 5-step generation. **2) Architecture:** We introduce a lightweight, training-efficient SideNet that grants bidirectional temporal perception to the off-the-shelf backbone. **3) Performance:** We validate BA-solver on ImageNet-$256^2$ and -$512^2$, demonstrating superior generation quality compared to state-of-the-art solvers and greater training efficiency over one- or few-step methods.

## 2. Related Work

**Training-free ODE Solvers.** Traditional ODE solvers, such as Euler, Heun, and RK45, typically require a large number of sequential NFEs for high-fidelity sampling, which imposes a significant bottleneck on inference speed. While recent specialized solvers like DPM-solvers (Lu et al., 2022; 2025; Xie et al., 2024), UniPC (Zhao et al., 2023), and DEIS (Zhang & Chen, 2022) have mitigated this issue by enabling high-fidelity generation with 20 NFEs in Diffusion Probabilistic Models, they still struggle to maintain sample quality in the extreme few-step generation (*e.g.*, 5 NFEs).

**One- or Few-Step Generation.** Recent research has increasingly focused on achieving high-quality generation in just one or a few steps. Prominent methods include Short-Cut (Frans et al., 2024), MeanFlow (Geng et al., 2025a;b), AYF (Sabour et al., 2025), pi-flow (Chen et al., 2025), FreeFlow (Tong et al., 2025), AlphaFlow (Zhang et al., 2025), and Consistency Models (Song et al., 2023; Song & Dhariwal, 2023). However, these approaches typically necessitate prohibitive training or distillation costs and lack the flexibility to be directly applied to off-the-shelf models.

**Efficient Fine-tuning.** Fine-tuning large backbones incurs high computational costs and memory demands. To address this, Parameter-Efficient Fine-Tuning (PEFT) (Han et al., 2024), such as LoRA (Hu et al., 2022), VPT (Jia et al., 2022), and Side-Tuning (Zhang et al., 2020; Sung et al., 2022), have been proposed. Among these, Side-Tuning is particularly notable for being both parameter- and memory-efficient. By introducing a lightweight SideNet isolated from the large backbone, it ensures that gradients do not backpropagate through the backbone during training. BA-solver leverages this lightweight SideNet as a plug-in module to endow the frozen backbone with bidirectional temporal perception.

## 3. Methodology

### 3.1. Preliminaries

**Flow Matching ODEs.** Continuous-time generative models formulate the generation process as a transport map from a simple prior distribution $p_1(\boldsymbol{x}) = \mathcal{N}(\boldsymbol{0}, \boldsymbol{I})$ to the complex data distribution $p_0(\boldsymbol{x})$. This transport is governed by an ODE. Flow matching directly regresses a velocity field $\boldsymbol{v}_\theta(\boldsymbol{x}_t, t)$ constructed to generate a specific probability path $p_t(\boldsymbol{x})$ interpolating between $p_1$ and $p_0$. The generation process is governed by the ODE:

$$\mathrm{d}\boldsymbol{x}_t = \boldsymbol{v}_\theta(\boldsymbol{x}_t, t)\mathrm{d}t, \quad t \in [1, 0]. \quad (1)$$

Discretizing the continuous time into intervals and solving Eq. (1) over interval $h$, we can express the transition using:

$$\boldsymbol{x}_{t-h} = \boldsymbol{x}_t - \int_{t-h}^{t} \boldsymbol{v}_\theta(\boldsymbol{x}_\tau, \tau)\mathrm{d}\tau. \quad (2)$$

**Extrapolation Solvers.** They utilize the current velocity $\boldsymbol{v}_t$, optionally incorporating historical states (*e.g.*, $\boldsymbol{v}_{t+h}, \boldsymbol{x}_{t+h}$) to estimate $\boldsymbol{x}_{t-h}$. The standard Euler solver (Stoer et al., 1980) is the simplest extrapolation solver, using only $\boldsymbol{v}_t$ to approximate the integral in Eq. (2):

$$\boldsymbol{x}_{t-h} = \boldsymbol{x}_t - h \cdot \boldsymbol{v}_\theta(\boldsymbol{x}_t, t). \quad (3)$$

It requires only one NFE to calculate $\boldsymbol{v}_t$ in each interval. However, this method often suffers from a local truncation error of $\mathcal{O}(h^2)$ and a global error of $\mathcal{O}(h)$. Consequently, fine-grained interval sizes are necessary to ensure accuracy, leading to high total sampling latency. Higher-order extrapolation solvers, such as Flow-DPM-solver (Xie et al., 2024), additionally utilize historical states. While it achieves higher accuracy compared to Euler solver, large extrapolation errors inherently limit the use of larger interval sizes, typically requiring at least 20 NFEs to achieve lossless sampling.

**Interpolation Solvers.** They utilize multiple velocities (*i.e.*, current velocity and intermediate velocities) within an interval to estimate $\boldsymbol{x}_{t-h}$. Heun solver (Stoer et al., 1980), a typical interpolation solver, approximates the integral using the average velocity at the start ($t$) and the estimated terminal ($t - h$) of the interval:

$$\boldsymbol{x}_{t-h} = \boldsymbol{x}_t - \frac{h}{2}\left[\boldsymbol{v}_\theta(\boldsymbol{x}_t, t) + \boldsymbol{v}_\theta(\tilde{\boldsymbol{x}}_{t-h}, t-h)\right], \quad (4)$$

where $\tilde{\boldsymbol{x}}_{t-h}$ is an initial estimate from an Euler step (*i.e.*, $\tilde{\boldsymbol{x}}_{t-h} = \boldsymbol{x}_t - h \cdot \boldsymbol{v}_\theta(\boldsymbol{x}_t, t)$). While interpolation solvers generally offer higher accuracy than extrapolation solvers, they require *sequential* multiple NFEs (*e.g.*, 2 for Heun) within a single interval, rather than parallelizable predictions. This sequential dependency causes a linear increase in latency per step, offsetting the efficiency gains with fewer intervals.

### 3.2. Proposed Approach: BA-solver

Based on the analysis in Sec. 3.1, we identify two critical factors governing sampling efficiency for high-quality generation: *1) the estimation accuracy of the integral term* in

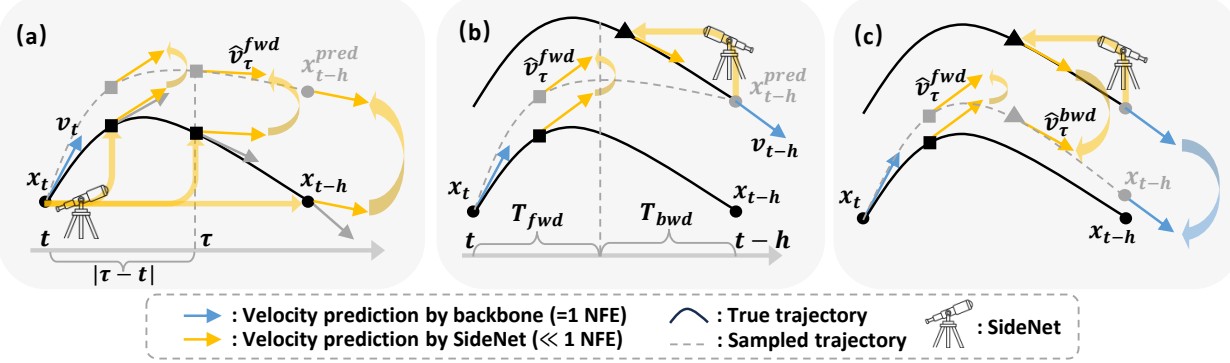

*Figure 3.* **(a) Forward Probe** using Single-Anchor Interpolation Solver to acquire $x_{t-h}^{pred}$. **(b) Backward Refinement** for intermediate velocities utilizing SideNet's lookback ability anchored on velocity $v_{t-h}$. **(c) Integration & State Reuse.** By high-order integration for two anchor velocities and multiple velocities, we can acquire a more accurate $x_{t-h}^{pred}$. Anchor $v_{t-h}$ is cached for reuse in the next interval.

Eq. (2); and *2) the number of NFEs required per interval.* Factor (1) dictates the total number of intervals. Specifically, higher estimation accuracy allows for larger interval sizes, thereby reducing the required intervals. Factor (2) determines the time latency per interval, as sequential NFEs involve repeated passes through FM model, significantly inflating latency. Consequently, our primary objective is to *enhance the precision of the integral estimation without incurring additional NFEs within each interval.*

### 3.2.1. BIDIRECTIONAL TEMPORAL PERCEPTION

To achieve this objective, we first introduce a lightweight SideNet, denoted as $\mathcal{S}_\phi$ and parameterized by $\phi$. In practice, the model size of $\mathcal{S}_\phi$ is approximately 0.03%–1.0% that of the backbone $v_\theta$. While $v_\theta$ is tasked with learning the global vector field, $\mathcal{S}_\phi$ is explicitly designed to learn bidirectional temporal perception. Specifically, given the current state $x_t$, the current velocity $v_t = v_\theta(x_t, t)$, and a bidirectional offset $\Delta t$, the SideNet predicts the velocity deviation at the time step $t + \Delta t$:

$$\begin{aligned}\hat{v}_{t+\Delta t} &= v_\theta(x_{t+\Delta t}, t + \Delta t) \\ &= v_t + \Delta t \cdot \mathcal{S}_\phi(x_t, v_t, t, \Delta t),\end{aligned} \quad (5)$$

where $\mathcal{S}_\phi$ predicts the first-order velocity deviation. Notably, this formulation ensures that $\hat{v}_{t+\Delta t}$ is equal to $v_t$ when the offset $\Delta t$ is zero. This guarantees that the performance of the combined model (*i.e.*, standard FM model + SideNet) is lower-bounded by that of the backbone. Assuming we have already acquired such SideNet $\mathcal{S}_\phi$ (training process will be discussed later in Sec. 3.2.3). In the following, we introduce BA-solver and its utilization of this temporal perception.

### 3.2.2. BI-ANCHOR VELOCITY INTEGRATION

**Observation:** *Single-anchor velocity interpolation is suboptimal[2].* Leveraging $\mathcal{S}_\phi$ anchored on only the current velocity $v_t$ (single-anchor), we can straightforwardly formulate an approximation of the integral in Eq. (2). By applying a

quadrature rule with $K$ nodes $\mathcal{T} = \{\tau_i\}_{i=1}^K$ and corresponding weights $\{\omega_i\}_{i=1}^K$ to approximation, we obtain:

$$x_{t-h} \approx x_t - (h \cdot v_t + \Delta_\mathcal{S}), \quad (6)$$

where the term $\Delta_\mathcal{S}$ is defined as:

$$\Delta_\mathcal{S} = \sum_{i=1}^K \omega_i \cdot (\tau_i - t) \cdot \mathcal{S}_\phi(x_t, v_t, t, \tau_i - t). \quad (7)$$

Every term in $\Delta_\mathcal{S}$ depends solely on the SideNet and can be computed in a **single parallelized batch**. Consequently, once the velocity $v_t$ is obtained from the backbone (1 NFE), the integral approximation incurs negligible computational cost ($\ll$ 1 NFE). Theoretically, since numerical integration with multiple intermediate velocities yields high-accuracy results for $x_{t-h}$[2], this scheme should support significantly larger interval sizes. However, we observe that the prediction fidelity of $\mathcal{S}_\phi$ inevitably degrades as the offset $|\tau - t|$ increases (*cf.*, Figure 3(a)), which can affect the estimation accuracy of the integral, thereby precluding the use of significantly larger interval sizes. This limitation highlights the necessity for a mechanism to reduce the impact of $\mathcal{S}_\phi$ prediction errors, motivating our proposed *Bi-Anchor Velocity Interpolation* strategy.

**Bi-Anchor Interpolation Solver.** To mitigate the fidelity degradation of $\mathcal{S}_\phi$ at larger offsets $|\tau - t|$, BA-solver fundamentally evolves the integration paradigm from the aforementioned single-anchor interpolation to the bi-anchor interpolation. Instead of relying on the single anchor velocity $v_t$, BA-solver utilizes the backbone to establish another anchor velocity at the terminal $(t - h)$ of the interval. Within this bounded interval, the SideNet functions as a bridge, approximating intermediate velocities from two directions anchored on $v_t$ and $v_{t-h}$ respectively. Specifically, BA-solver proceeds in three distinct phases within each interval:

*(1) Forward Probe.* As shown in Figure 3(a), given the

---

[2] The detailed derivation is left in the appendix.

---

**Algorithm 1** BA-solver Sampling

**Require:** Backbone $v_\theta$, SideNet $\mathcal{S}_\phi$, Noise $x_1$, Steps $N$.
**Ensure:** Generated sample $x_0$.
1: $h \leftarrow 1/N, t \leftarrow 1, v_t \leftarrow v_\theta(x_1, t)$
2: **for** $i = 0$ **to** $N - 1$ **do**
3:     $\mathcal{T} \leftarrow$ Quadrature Nodes in $(t - h, t)$
4:     *// Phase 1: Forward Probe*
5:     $\hat{v}_\tau^{fwd} \leftarrow v_t + \Delta_\mathcal{T} \cdot \mathcal{S}_\phi(x_t, v_t, t, \Delta_\mathcal{T}), \quad \forall \tau \in \mathcal{T}$
6:     $x_{t-h}^{pred} \leftarrow x_t - h \cdot \text{Quadrature}(\{\hat{v}_\tau^{fwd}\})$
7:     **if** $i = N - 1$ **then**
8:         **return** $x_{t-h}^{pred}$
9:     **end if**
10:    *// Phase 2: Backward Refinement*
11:    $v_{t-h} \leftarrow v_\theta(x_{t-h}^{pred}, t - h)$
12:    $\mathcal{T}_{bwd} \leftarrow \{\tau \in \mathcal{T} \mid |\tau - (t - h)| < |\tau - t|\}$
13:    $\hat{v}_\tau^{bwd} \leftarrow v_{t-h} + \Delta_{\mathcal{T}_{bwd}} \cdot \mathcal{S}_\phi(x_{t-h}^{pred}, v_{t-h}, t - h, \Delta_{\mathcal{T}_{bwd}}), \quad \forall \tau \in \mathcal{T}_{bwd}$
14:    *// Phase 3: Integration & State Reuse*
15:    $\mathcal{V}^* \leftarrow \{\hat{v}_\tau^{fwd} \mid \tau \in \mathcal{T}_{fwd}\} \cup \{\hat{v}_\tau^{bwd} \mid \tau \in \mathcal{T}_{bwd}\}$
16:    $x_{t-h} \leftarrow x_t - h \cdot \text{Quadrature}(\mathcal{V}^*)$
17:    $x_t \leftarrow x_{t-h}, t \leftarrow t - h$
18:    $v_t \leftarrow v_{t-h}$ {Reuse for next interval}
19: **end for**
20: **return** $x_0$

---

**Algorithm 2** BA-solver Chain-based Training

**Require:** Backbone $v_\theta$, SideNet $\mathcal{S}_\phi$, Data $x_{data}$, Chain Length $K$.
**Ensure:** Optimized parameters $\phi$.
1: Sample $t \sim \mathcal{U}[0, 1], x_{noise} \sim \mathcal{N}(\mathbf{0}, \mathbf{I})$
2: $x_t \leftarrow (1 - t)x_{data} + t x_{noise}$
3: $v_t \leftarrow v_\theta(x_t, t), \mathcal{L}_{total} \leftarrow 0$
4: **for** $k = 1$ **to** $K$ **do**
5:    Sample interval size $h$ from a truncated exponential distribution and offsets $\Delta_\mathcal{T}$ for Quadrature
6:    *// Phase 1: Solver Simulation*
7:    $\hat{v}_\tau \leftarrow v_t + \Delta_\mathcal{T} \cdot \mathcal{S}_\phi(x_t, v_t, t, \Delta_\mathcal{T}), \quad \forall \tau \in \mathcal{T}$
8:    $x_{t-h} \leftarrow x_t - h \cdot \text{Quadrature}(\{\hat{v}_\tau\})$
9:    *// Phase 2: Velocity Matching*
10:    $v_{t-h} \leftarrow v_\theta(x_{t-h}, t - h)$
11:    $v_{t-h}^{pred} \leftarrow v_t - h \cdot \mathcal{S}_\phi(x_t, v_t, t, -h)$
12:    $\mathcal{L}_{total} \leftarrow \mathcal{L}_{total} + \|v_{t-h}^{pred} - \text{SG}(v_{t-h})\|_2^2$
13:    $x_t \leftarrow x_{t-h}, t \leftarrow t - h$
14:    $v_t \leftarrow v_{t-h}$ {Reuse for next interval}
15: **end for**
16: **Backward** $\nabla_\phi(\mathcal{L}_{total}/K)$

---

anchor $v_t$ (inherited from the previous step), we first employ the SideNet to simultaneously predict the intermediate velocities for quadrature nodes $\mathcal{T}$. This yields a set of provisional velocity vectors $\{\hat{v}_\tau^{fwd}\}_{\tau \in \mathcal{T}}$ anchored on $v_t$. Using these estimates, we compute the predicted next state $x_{t-h}^{pred}$:

$$x_{t-h}^{pred} = x_t - h \cdot \text{Quadrature}(\{\hat{v}_\tau^{fwd}\}), \quad (8)$$

where Quadrature$(\cdot)$ denotes a generic fixed high-order numerical integration rule (*e.g.*, 4-point Gauss-Lobatto quadrature (Stoer et al., 1980)).

*(2) Backward Refinement.* As consistent with our prior analysis, the single-anchor mechanism is insufficient to enable significantly larger interval sizes, as the forward prediction $\hat{v}_\tau^{fwd}$ becomes increasingly unreliable at large offsets. To address this, we obtain terminal velocity $v_{t-h} = v_\theta(x_{t-h}^{pred}, t - h)$ to establish another anchor velocity (*cf.*, Figure 3(b)). This allows us to perform a proximity-based partition of $\mathcal{T}$: we define the backward set $\mathcal{T}_{bwd} = \{\tau \in \mathcal{T} \mid |\tau - (t - h)| < |\tau - t|\}$ containing nodes closer to terminal of the interval, and the forward set $\mathcal{T}_{fwd} = \mathcal{T} \setminus \mathcal{T}_{bwd}$. Facilitated by bidirectional perception of SideNet, the anchor $v_{t-h}$ enables a backward lookback, generating refined velocities $\{\hat{v}_\tau^{bwd}\}$. Consequently, for all $\tau \in \mathcal{T}_{bwd}$, we substitute the error-prone forward predictions $\hat{v}_\tau^{fwd}$ with their backward counterparts $\hat{v}_\tau^{bwd}$. Since the offset from the terminal is significantly reduced for these nodes, this refinement strategy effectively minimizes the

approximation error for intermediate velocities.

*(3) Integration & State Reuse.* Finally, we perform the corrector update by fusing the most reliable velocities available to acquire $x_{t-h}$ (*cf.*, Figure 3(c)): proximity-based predictions for the intermediate velocities (forward predictions for $\mathcal{T}_{fwd}$ and backward refinements for $\mathcal{T}_{bwd}$). The final state update is computed using a high-order quadrature rule:

$$x_{t-h} = x_t - h \cdot \text{Quadrature}(\mathcal{V}^*), \quad (9)$$

where $\mathcal{V}^*$ aggregates the proximity-based predictions:

$$\mathcal{V}^* = \{\hat{v}_\tau^{fwd} \mid \tau \in \mathcal{T}_{fwd}\} \cup \{\hat{v}_\tau^{bwd} \mid \tau \in \mathcal{T}_{bwd}\}. \quad (10)$$

To achieve optimal efficiency, we implement a *State Reuse* mechanism: the anchor $v_{t-h}$ computed in the current interval is cached and reused as the start anchor $v_{t-h}$ for the subsequent interval. Consequently, the expensive backbone evaluation is performed only once per interval, ensuring that BA-solver maintains a strict Exact-N NFE cost (where $N$ is the number of intervals), matching the latency of extrapolation solvers while delivering a high-accuracy approximation characteristic of interpolation solvers. The detailed sampling process is shown in Algorithm 1.

### 3.2.3. CHAIN-BASED TRAINING FOR SIDENET

To ensure robustness during inference, we employ a $K$-interval chain training strategy that mimics the actual sequential generation process. As outlined in Algorithm 2, each iteration consists of two distinct phases: solver simulation and velocity matching.

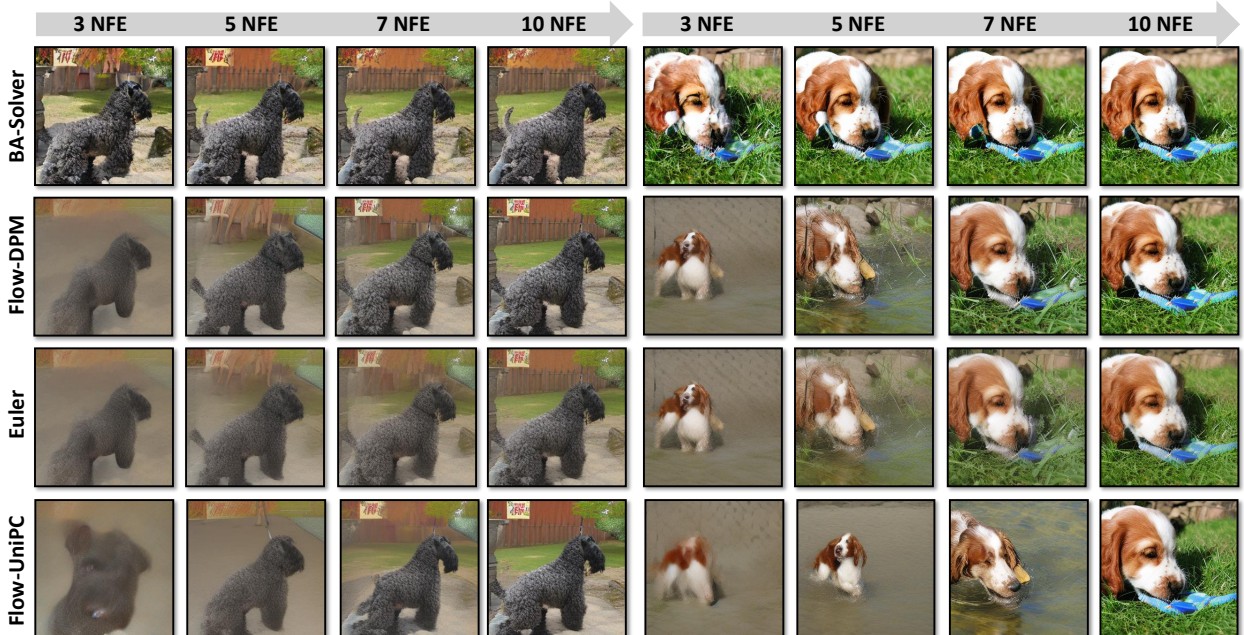

*Figure 4.* **Qualitative comparison of generated samples across different solvers and NFEs.** We visualize random samples generated by our BA-solver and baseline methods (Flow-DPM, Euler, and Flow-UniPC solvers) at 3, 5, 7, and 10 NFEs.

In the simulation phase, by sampling $h$ from a truncated exponential distribution and using SideNet to estimate intermediate velocities, we advance the state from $\boldsymbol{x}_t$ to $\boldsymbol{x}_{t-h}$ via high-order quadrature. This step effectively simulates the solver's trajectory, exposing the model to realistic error accumulation patterns that occur during inference. Subsequently, in the matching phase, we optimize parameters $\phi$ by minimizing the discrepancy between SideNet's velocity prediction and the ground truth derived from the backbone:

$$\mathcal{L}_\phi = \mathbb{E}_{t,\boldsymbol{x}_t,h} \left\| \underbrace{[\boldsymbol{v}_t - h \cdot \mathcal{S}_\phi(\boldsymbol{x}_t, \boldsymbol{v}_t, t, -h)]}_{\boldsymbol{v}_{t-h}^{\text{pred}}} - \text{SG}(\boldsymbol{v}_{t-h}) \right\|_2^2,$$

(11)

where $\boldsymbol{v}_{t-h} = \boldsymbol{v}_\theta(\boldsymbol{x}_{t-h}, t - h)$ represents the target velocity at the new state, and $\text{SG}(\cdot)$ denotes the Stop Gradient operator. The use of stop gradient stabilizes training by preventing the SideNet from modifying the backbone's targets. Crucially, to maintain high training throughput, the target $\boldsymbol{v}_{t-h}$ computed in step $k$ is directly reused as the starting velocity anchor $\boldsymbol{v}_t$ for step $k+1$. Consequently, a length-$K$ chain requires only $K+1$ backbone evaluations in total, ensuring that the training cost remains minimal even with multi-step unrolling.

## 4. Experiments

### 4.1. Experimental Setting

**Baselines and Datasets.** We benchmarked BA-solver against a comprehensive set of baselines, comprising four training-free solvers (**Euler**, **Heun**, **Flow-DPM Solver** (Xie

et al., 2024), and **Flow-UniPC Solver** (Zhao et al., 2023)) and four training-based one- or few-step generation methods (**iCT** (Song & Dhariwal, 2023), **Shortcut** (Frans et al., 2024), **MeanFlow** (Geng et al., 2025a)[3], $\alpha$-**flow** (Cheng et al., 2025), and $\pi$-**Flow** (Chen et al., 2025)). We employed the REPA-enhanced (Yu et al., 2025) SiT (Ma et al., 2024) as the backbone for experiments on ImageNet-256[2] and -512[2] (Deng et al., 2009).

**Evaluation Setting and Metrics.** To evaluate performance against training-free solvers, we employed Frechet Inception Distance (FID) (Heusel et al., 2017), sliding window Fréchet Inception Distance (sFID) (Nash et al., 2021), Inception Score (IS), and Precision-Recall (Kynkäänniemi et al., 2019) across different NFE numbers. A consistent CFG (Ho & Salimans, 2022) interval (Kynkäänniemi et al., 2024) setting was maintained across all methods to ensure fair comparison. When benchmarking against training-based one- or few-step methods, we further assessed training efficiency by comparing the required training iterations (normalized to a batch size of $4,096$) and the number of trainable parameters.

**Implementation Details.** We utilize 3-point Gauss-Legendre quadrature (Stoer et al., 1980) during training and 4-point Gauss-Lobatto quadrature for sampling. The detailed training settings are provided in the appendix. In the sampling phase, the SideNet predicts two intermediate velocities within each interval (the other velocities are anchor velocities provided by the backbone). We also investigate the impact of different numbers of intermediate velocities,

---

[3]We utilize an open-source community implementation.

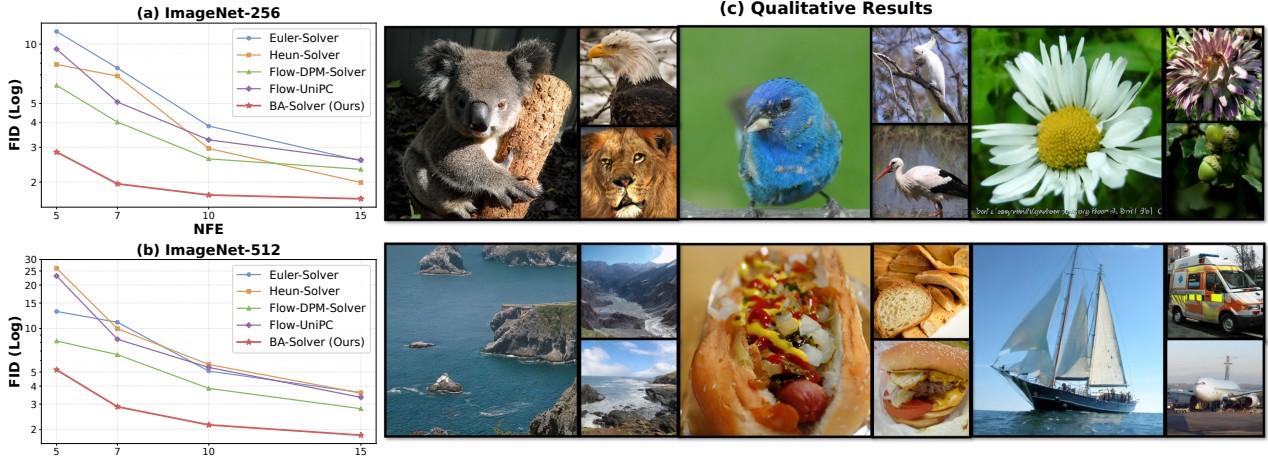

*Figure 5.* **Performance comparison on ImageNet.** **(a)-(b)** FID scores varying with NFE on ImageNet-$256^2$ and ImageNet-$512^2$. BA-Solver achieves superior performance compared to baselines. **(c)** Visual samples of BA-Solver on ImageNet-$512^2$ with 7 NFEs.

*Table 1.* **Quantitative comparison on class-conditional ImageNet-**$256^2$ **and ImageNet-**$512^2$**.** We compare BA-solver against other training-free solvers under NFE=15, 7 and 5. ↑/↓ indicates higher/lower is better. Best results in each block are marked in **bold**.

| Methods | ImageNet-$256^2$ | | | | | ImageNet-$512^2$ | | | | |
|---|---|---|---|---|---|---|---|---|---|---|
| | IS ↑ | FID ↓ | sFID ↓ | Prec. ↑ | Rec. ↑ | IS ↑ | FID ↓ | sFID ↓ | Prec. ↑ | Rec. ↑ |
| *Number of Function Evaluations (NFE) = 5* | | | | | | | | | | |
| Euler-Solver (Stoer et al., 1980) | 214.59 | 11.58 | 12.01 | 0.66 | 0.48 | 203.20 | 13.12 | 15.20 | 0.71 | 0.44 |
| Heun-Solver (Stoer et al., 1980) | 238.31 | 7.88 | **7.04** | 0.70 | 0.51 | 114.00 | 26.11 | 17.97 | 0.59 | 0.53 |
| Flow-UniPC (Zhao et al., 2023) | 243.64 | 9.43 | 10.85 | 0.70 | 0.42 | 127.42 | 23.08 | 18.23 | 0.62 | 0.48 |
| Flow-DPM (Xie et al., 2024) | 261.90 | 6.18 | 7.04 | 0.73 | 0.52 | 256.66 | 8.20 | **10.67** | 0.76 | 0.47 |
| **BA-solver (Ours)** | **306.22** | **2.84** | 8.09 | **0.76** | **0.63** | **270.26** | **5.18** | 12.45 | **0.78** | **0.60** |
| *Number of Function Evaluations (NFE) = 7* | | | | | | | | | | |
| Euler-Solver (Stoer et al., 1980) | 238.54 | 7.56 | 9.09 | 0.71 | 0.55 | 204.68 | 11.04 | 13.17 | 0.73 | 0.52 |
| Heun-Solver (Stoer et al., 1980) | 220.51 | 6.89 | 7.72 | 0.70 | 0.60 | 214.10 | 9.98 | 11.17 | 0.73 | 0.54 |
| Flow-UniPC (Zhao et al., 2023) | 258.89 | 5.09 | 6.12 | 0.74 | 0.57 | 243.45 | 8.43 | 9.20 | 0.75 | 0.49 |
| Flow-DPM (Xie et al., 2024) | 280.16 | 4.03 | 7.74 | 0.76 | 0.58 | 255.34 | 6.59 | 10.99 | 0.77 | 0.54 |
| **BA-solver (Ours)** | **320.49** | **1.96** | **6.06** | **0.78** | **0.64** | **300.79** | **2.88** | **8.35** | **0.80** | **0.61** |
| *Number of Function Evaluations (NFE) = 15* | | | | | | | | | | |
| Euler-Solver (Stoer et al., 1980) | 326.91 | 2.56 | 4.76 | 0.82 | 0.57 | 266.41 | 3.61 | 8.57 | 0.80 | 0.57 |
| Heun-Solver (Stoer et al., 1980) | 314.36 | 1.99 | 5.42 | 0.81 | 0.61 | 250.30 | 3.58 | 7.82 | 0.79 | 0.61 |
| Flow-UniPC (Zhao et al., 2023) | 327.72 | 2.59 | 8.37 | 0.82 | 0.58 | 272.55 | 3.34 | 7.99 | 0.80 | 0.57 |
| Flow-DPM (Xie et al., 2024) | **341.85** | 2.32 | 5.49 | **0.83** | 0.58 | **287.26** | 2.79 | 7.79 | **0.80** | 0.58 |
| **BA-solver (Ours)** | 326.67 | **1.65** | **4.43** | 0.81 | **0.62** | 272.14 | **1.83** | **5.04** | 0.80 | **0.63** |

the quadrature rules, and the anchor mechanism. These comparisons are provided in the ablation study.

### 4.2. Main Results

**Comparison with Solver Methods.** As shown in Table 1. BA-solver demonstrates significant advantages over training-free solvers, particularly in the few-step regime. On ImageNet-$256^2$, BA-solver surpasses other baselines in most of the metrics. Specifically, it achieves a state-of-the-art FID of **1.96** at just 7 NFEs, substantially outperforming Flow-DPM (4.03) and Euler (7.56). This superiority extends to ImageNet-$512^2$, where our method attains an FID of **2.88** at 7 NFEs, consistently surpassing baselines. Additionally, our convergence analysis highlights the exceptional sample efficiency of our method in Figure 5(a)-(b). BA-solver converges rapidly, achieving the highest generation quality at 15 NFEs (FID 1.72) on ImageNet-$256^2$.

We provide qualitative comparison of generated samples across baselines under different NFEs in Figure 4. At extremely low NFEs (3-5 NFEs), BA-solver produces structurally coherent and sharp images. In contrast, baselines often exhibit noticeable artifacts and blurriness, confirming BA-solver's effectiveness in mitigating discretization errors even in extremely low NFEs. Qualitative results on ImageNet-$512^2$ in Figure 5(c) demonstrate the scalability of our method. BA-solver maintains global coherence and rich textural details at higher 512 resolutions with only 7 NFEs.

**Comparison with One- or Few-step Generation Methods.** Table 2 highlights the superior efficiency of our proposed BA-solver compared to state-of-the-art one- or few-step generation methods. Notably, BA-solver demonstrates unprecedented training efficiency, converging in just 250 iterations (with a batch size of 4,096). This stands in stark contrast to competitors such as iCT and MeanFlow, which typically

*Table 2.* **Efficiency Comparison on class-conditional ImageNet-$256^2$.** We compare training cost (iterations and trainable parameters) and generation quality (FID) with state-of-the-art training-based one- or few-step methods. (M) indicates Million Parameters.

| Method | FID ↓ | Iterations | Training(M) | NFE |
|---|---|---|---|---|
| Shortcut (Frans et al., 2024) | 10.60 | 78,000 | 675 | 1 |
| MeanFlow (DiT) (Geng et al., 2025a) | 3.43 | 313,000 | 676 | 1 |
| $\pi$-Flow (Chen et al., 2025) | 2.85 | 140,000 | 676 | 1 |
| $\alpha$-flow (Cheng et al., 2025) | 2.58 | 95,000 | 676 | 1 |
| iCT (Song & Dhariwal, 2023) | 20.30 | 800,000 | 675 | 2 |
| MeanFlow (DiT) (Geng et al., 2025a) | 2.20 | 313,000 | 676 | 2 |
| MeanFlow (SiT)[3] (Geng et al., 2025a) | 2.55 | 41,000 | 675 | 2 |
| $\pi$-Flow (Chen et al., 2025) | 1.97 | 24,000 | 676 | 2 |
| $\alpha$-flow (Cheng et al., 2025) | 2.15 | 95,000 | 676 | 2 |
| MeanFlow (SiT)[3] (Geng et al., 2025a) | 5.58 | 41,000 | 675 | 8 |
| **BA-solver (Ours)** | **1.85** | **250** | **6** | 8 |

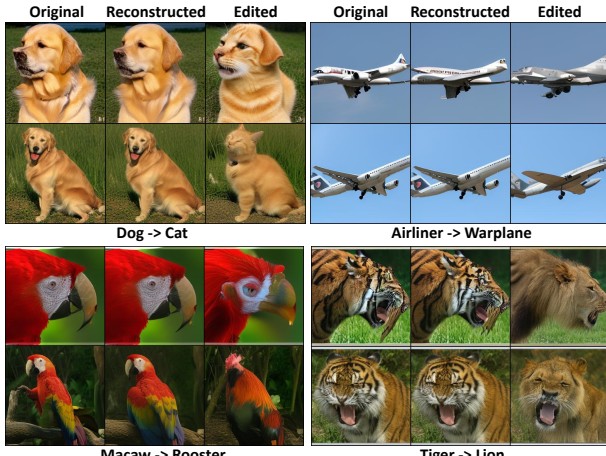

*Figure 6.* **Qualitative results of image editing using BA-solver.** We demonstrate that BA-solver can perform high-quality editing on class-conditional ImageNet-$256^2$ within only 10 NFEs.

require hundreds of thousands of iterations (*e.g.*, 313K–800K) to achieve comparable FID. Furthermore, BA-solver is highly parameter-efficient. By freezing the heavy backbone, it requires optimizing only 6M parameters, whereas other methods often necessitate fine-tuning the entire model (over 675M parameters). Despite these orders-of-magnitude reductions in both training iterations and trainable parameters, BA-solver achieves a state-of-the-art FID of **1.85** at 8 NFE. Additionally, while prior methods are often rigidly optimized for specific low-step regimes and fail to scale with increased NFEs (*e.g.*, MeanFlow's FID degrades to 5.58 with 5 NFE due to fixed CFG scale during training), BA-solver offers a robust, low-cost solution that excels in both training efficiency and generation quality. A detailed version of Table 2 is in the appendix.

**BA-solver for Image Editing.** BA-solver facilitates high-quality image editing by leveraging the reversibility of FM ODEs. As shown in Figure 6, BA-solver successfully transforms various object classes on class-conditional ImageNet-$256^2$ in 10 NFEs (*e.g.*, changing a "Dog" to a "Cat"). In contrast, most existing training-based one- or few-step generation methods lack this capability due to disruption of the

*Table 3.* **Ablation study on Intermediate Velocities, Anchor Mechanism, and Quadrature Rules.** We compare the impact of intermediate velocities number ($K$), anchor strategies, and integration rules under 7 NFEs on ImageNet-$256^2$. ↓ indicates lower is better, ↑ indicates higher is better.

| $K$ | Anchor | Rule | FID ↓ | sFID ↓ | IS ↑ | Rec. ↑ | Prec. ↑ |
|---|---|---|---|---|---|---|---|
| 1 | Bi-Anchor | Gauss-Lobatto | **1.89** | **5.84** | 322.21 | 0.6413 | 0.7785 |
| 2 | Bi-Anchor | Gauss-Lobatto | 1.96 | 6.06 | 320.49 | 0.6375 | **0.7823** |
| 2 | Bi-Anchor | Simpson | 1.97 | 6.17 | 321.07 | 0.6400 | 0.7776 |
| 2 | Single-Anchor | Gauss-Lobatto | 4.35 | 9.59 | 256.52 | **0.6567** | 0.7230 |
| 0 | Bi-Anchor | Gauss-Lobatto | 3.54 | 8.06 | 281.19 | 0.6089 | 0.7619 |

FM ODEs' mathematical structure.

## 5. Ablation Study

As shown in Table 3, we evaluate the contributions of our key components on ImageNet-$256^2$.

**Impact of Anchor Mechanism.** Switching from our Bi-Anchor strategy to a Single-Anchor approach results in a significant performance drop (FID degrades from 1.96 to 4.35). This highlights the necessity of the bi-anchor mechanism, which leverages both the starting point and the predicted endpoint for effective error correction.

**Impact of Quadrature Rules.** Comparing integration schemes with $K = 2$ intermediate nodes, the Gauss-Lobatto rule achieves slightly better performance than Simpson's rule (FID 1.96 vs. 1.97). This suggests that while the specific choice of quadrature weights has a marginal effect, the Gauss-Lobatto node placement (pushing nodes towards the boundaries) offers optimal stability for our solver.

**Effect of Intermediate Velocities.** Eliminating intermediate velocities entirely (i.e., setting $K = 0$) degrades the FID to 3.54, validating the critical role of the side network's predictions. While $K = 1$ yields the highest generation quality (representing a special case of $K = 2$ with backward refinement on a single midpoint), we select $K = 2$ as our default configuration. We hypothesize that concentrating backward refinement on multiple nodes is more effective than distributing it without any nodes, particularly in low-NFE regimes. We choose $K = 2$ to leverage its theoretically superior integration precision.

## 6. Conclusion

In this paper, we presented the **Bi-Anchor Interpolation Solver (BA-solver)**, a novel framework that effectively bridges the gap between computationally expensive training-free solvers and resource-intensive training-based acceleration methods. By introducing a lightweight SideNet, we endow frozen flow matching backbones with bidirectional temporal perception, enabling high-order efficient numerical integration. BA-solver achieves state-of-the-art generation quality with as few as 5 to 10 NFEs, matching the performance of standard Euler solvers requiring 100+ NFEs.

## Acknowledgment

This work was supported by National Natural Science Foundation of China (NSFC) Young Scientists Fund Category B (62522216), National Natural Science Foundation of China (NSFC) Young Scientists Fund Category C (62402408), Hong Kong SAR Research Grants Council (RGC) Early Career Scheme (26208924), and Hong Kong SAR Research Grants Council (RGC) General Research Fund (16219025).

## Impact Statement

This paper presents a novel acceleration method, the Bi-Anchor Interpolation Solver (BA-Solver), designed to enhance the sampling efficiency of flow matching and diffusion models. By significantly reducing the Number of Function Evaluations (NFEs) required for high-quality generation, our work contributes positively to the goal of "Green AI". The reduction in computational cost directly translates to lower energy consumption and a reduced carbon footprint for large-scale generative tasks. Furthermore, the improved efficiency lowers the hardware barrier, facilitating the deployment of advanced generative models on edge devices and making the technology more accessible to a broader range of users.

However, we acknowledge that accelerating generative modeling inherently carries dual-use risks. The ability to generate high-fidelity images rapidly and at a lower cost could be exploited by malicious actors to produce mass-scale disinformation, deepfakes, or spam content more efficiently. While our method focuses on the algorithmic efficiency of sampling rather than the content generation itself, the lowered barrier to entry underscores the urgent need for robust content authentication mechanisms, such as watermarking and deepfake detection systems. We advocate for the responsible disclosure and deployment of such acceleration techniques, ensuring that safety measures evolve in parallel with improvements in generative speed.

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

# A. Error Analysis of Single-Anchor Interpolation

In this section, we provide a formal analysis of the Local Truncation Error (LTE) for the single-anchor interpolation. We demonstrate that relying solely on the starting anchor $\boldsymbol{v}_t$ results in an error bound that grows quadratically with the interval size $h$, thereby justifying the necessity of the bi-anchor refinement mechanism proposed in the main text.

## A.1. Problem Formulation

Consider the FM ODE governing the generative process:

$$\mathrm{d}\boldsymbol{x}_t = \boldsymbol{v}_\theta(\boldsymbol{x}_t, t)\mathrm{d}t. \tag{12}$$

The true state at time $t - h$ is obtained via the exact integral:

$$\boldsymbol{x}_{t-h} = \boldsymbol{x}_t - \int_{t-h}^{t} \boldsymbol{v}_\theta(\boldsymbol{x}_\tau, \tau)\mathrm{d}\tau. \tag{13}$$

In the single-anchor scheme (corresponding to Phase 1: Forward Probe in our algorithm), we approximate the velocity at any intermediate time $\tau \in [t - h, t]$ using the SideNet anchored at the current state $(\boldsymbol{x}_t, \boldsymbol{v}_t)$. Let $\delta = \tau - t$ denote the time offset. The approximated velocity is defined as:

$$\hat{\boldsymbol{v}}_\tau^{\mathrm{SA}} = \boldsymbol{v}_t + \delta \cdot \mathcal{S}_\phi(\boldsymbol{x}_t, \boldsymbol{v}_t, t, \delta). \tag{14}$$

Consequently, the estimated state is given by:

$$\boldsymbol{x}_{t-h}^{\mathrm{SA}} = \boldsymbol{x}_t - \int_{t-h}^{t} \hat{\boldsymbol{v}}_\tau^{\mathrm{SA}}\mathrm{d}\tau. \tag{15}$$

### A.2. Error Bound Derivation

We define the LTE $\mathcal{E}(h)$ as the norm of the difference between the true state and the single-anchor approximation:

$$\mathcal{E}(h) = \|\boldsymbol{x}_{t-h} - \boldsymbol{x}_{t-h}^{\mathrm{SA}}\|. \tag{16}$$

Substituting the integral formulations into the error definition, we obtain:

$$\mathcal{E}(h) = \left\| \int_{t-h}^{t} \left( \boldsymbol{v}_\theta(\boldsymbol{x}_\tau, \tau) - \hat{\boldsymbol{v}}_\tau^{\mathrm{SA}} \right) \mathrm{d}\tau \right\|. \tag{17}$$

Applying the triangle inequality yields:

$$\mathcal{E}(h) \leq \int_{t-h}^{t} \left\| \boldsymbol{v}_\theta(\boldsymbol{x}_\tau, \tau) - \hat{\boldsymbol{v}}_\tau^{\mathrm{SA}} \right\| \mathrm{d}\tau. \tag{18}$$

To analyze the integrand $\epsilon(\tau) = \|\boldsymbol{v}_\theta(\boldsymbol{x}_\tau, \tau) - \hat{\boldsymbol{v}}_\tau^{\mathrm{SA}}\|$, we consider the first-order Taylor expansion of the true velocity field around $t$:

$$\boldsymbol{v}_\theta(\boldsymbol{x}_\tau, \tau) = \boldsymbol{v}_\theta(\boldsymbol{x}_t, t) + \nabla_t \boldsymbol{v}_\theta \cdot \delta + \nabla_{\boldsymbol{x}} \boldsymbol{v}_\theta \cdot (\boldsymbol{x}_\tau - \boldsymbol{x}_t) + \mathcal{O}(\delta^2). \tag{19}$$

The single-anchor estimator $\hat{\boldsymbol{v}}_\tau^{\mathrm{SA}}$ models the temporal evolution term $\nabla_t \boldsymbol{v} \cdot \delta$ via the SideNet term $\delta \cdot \mathcal{S}_\phi$. However, a critical source of error arises from the state drift term $\nabla_{\boldsymbol{x}} \boldsymbol{v} \cdot (\boldsymbol{x}_\tau - \boldsymbol{x}_t)$, which the SideNet cannot perceive effectively as it is conditioned solely on the fixed state $\boldsymbol{x}_t$.

Due to this discrepancy, a lightweight SideNet can efficiently capture temporal evolution, but only *partially* absorb the state drift. The uncaptured drift manifests as the dominant systematic error, which empirically acts as if governed by a smaller effective Lipschitz constant $L_{\mathrm{eff}}$ compared to the full vector field's Lipschitz constant $L$.

Assuming the SideNet has a bounded approximation error $\eta(\delta)$, we can bound the velocity error at offset $\delta$:

$$\|\boldsymbol{v}_\theta(\boldsymbol{x}_\tau, \tau) - \hat{\boldsymbol{v}}_\tau^{\mathrm{SA}}\| \leq \underbrace{\eta(\delta)}_{\text{Network Error}} + \underbrace{L_{\mathrm{eff}}\|\boldsymbol{x}_\tau - \boldsymbol{x}_t\|}_{\text{Uncaptured Drift}}. \tag{20}$$

For a standard solver, the state displacement is approximately linear with respect to time, i.e., $\|\boldsymbol{x}_\tau - \boldsymbol{x}_t\| \approx \|\boldsymbol{v}\| \cdot |\tau - t| = C \cdot |\delta|$, where $C$ is a constant related to the velocity magnitude. Thus, the error integrand grows linearly with the absolute offset $|\delta|$:

$$\epsilon(\tau) \approx \eta(\delta) + L_{\mathrm{eff}} \cdot C \cdot |\delta|. \tag{21}$$

Integrating this over the interval $[t - h, t]$ (noting that the length of integration is $h$):

$$\mathcal{E}(h) \leq \int_0^h (\eta(\delta) + L_{\mathrm{eff}} \cdot C \cdot \delta)\mathrm{d}\delta. \tag{22}$$

At larger offsets, the uncaptured state drift dominates the total approximation error. Treating the network's fitting error $\eta(\delta)$ as an empirically minor term in our setting, the total error can be approximated by the dominant systematic drift:

$$\mathcal{E}(h) \approx \frac{1}{2} L_{\mathrm{eff}} C h^2 = \mathcal{O}(h^2). \tag{23}$$

### A.3. Conclusion

The error bound derived above demonstrates that the single-anchor approximation suffers from a local truncation error that scales quadratically with the interval size $h$ ($\mathcal{O}(h^2)$). This error is primarily driven by the extrapolation drift $L_{\mathrm{eff}}\|\boldsymbol{x}_\tau - \boldsymbol{x}_t\|$, which accumulates as the offset $\delta$ increases. This theoretical result confirms the empirical observation in Figure 3(a) and motivates the proposed bi-anchor strategy. By introducing a second anchor $\boldsymbol{v}_{t-h}$, we effectively reset the offset $\delta$ near the interval boundary, thereby significantly mitigating the drift error.

## B. Error Analysis of Bi-Anchor Interpolation

Building upon the analysis of the single-anchor scheme, we now evaluate the error bound for the proposed bi-anchor interpolation. We demonstrate that by introducing a second anchor at the terminal state, we effectively halve the maximum offset, resulting in a largely reduced error bound.

### B.1. Bi-Anchor Formulation

In the bi-anchor scheme (corresponding to Phase 2 & 3 in our algorithm), we utilize two anchor velocities: the starting anchor $\boldsymbol{v}_t$ and the terminal anchor $\boldsymbol{v}_{t-h}$. The integration interval $[t - h, t]$ is partitioned into two sub-intervals based on proximity to these anchors. Let $t_{mid} = t - h/2$ be the midpoint. The approximated velocity $\hat{\boldsymbol{v}}_\tau^{\mathrm{BA}}$ is defined piecewise:

$$\hat{\boldsymbol{v}}_\tau^{\mathrm{BA}} = \begin{cases} \boldsymbol{v}_t + \delta \cdot \mathcal{S}_\phi(\boldsymbol{x}_t, \boldsymbol{v}_t, t, \delta), & \tau \in [t_{mid}, t] \\ \boldsymbol{v}_{t-h} + \delta' \cdot \mathcal{S}_\phi(\boldsymbol{x}_{t-h}, \boldsymbol{v}_{t-h}, \\ \qquad\qquad\qquad t - h, \delta'), & \tau \in [t - h, t_{mid}), \end{cases} \tag{24}$$

where $\delta = \tau - t$ is the offset from the start, and $\delta' = \tau - (t - h)$ is the offset from the terminal. Crucially, in both cases, the magnitude of the offset is bounded by half the interval size: $|\delta|, |\delta'| \leq h/2$.

### B.2. Reduced Error Bound Derivation

Similar to the single-anchor analysis, we define the error $\mathcal{E}_{\text{BA}}(h)$ as the norm of the difference between the true and estimated states. The total error can be decomposed into the sum of errors over the forward and backward sub-intervals:

$$\mathcal{E}_{\text{BA}}(h) \leq \underbrace{\int_{t_{mid}}^{t} \|\boldsymbol{v}_\theta - \hat{\boldsymbol{v}}_\tau^{\text{fwd}}\| \mathrm{d}\tau}_{\text{Forward Error}} + \underbrace{\int_{t-h}^{t_{mid}} \|\boldsymbol{v}_\theta - \hat{\boldsymbol{v}}_\tau^{\text{bwd}}\| \mathrm{d}\tau}_{\text{Backward Error}}$$

(25)

Using the same Lipschitz assumptions ($L_{\text{eff}}$) and state displacement linearity ($C$) as in Eq 22, the error density $\epsilon(\tau)$ is proportional to the distance from the nearest anchor. For the forward part, the offset is $|\tau - t|$, ranging from 0 to $h/2$. For the backward part, the offset is $|\tau - (t - h)|$, also ranging from 0 to $h/2$. Substituting the linear drift approximation $\epsilon(\delta) \approx L_{\text{eff}} \cdot C \cdot \delta$:

$$\begin{aligned}
\mathcal{E}_{\text{BA}}(h) &\leq \int_0^{h/2} (L_{\text{eff}} \cdot C \cdot \delta) \mathrm{d}\delta + \int_0^{h/2} (L_{\text{eff}} \cdot C \cdot \delta') \mathrm{d}\delta' \\
&= \left[ \frac{1}{2} L_{\text{eff}} C \delta^2 \right]_0^{h/2} + \left[ \frac{1}{2} L_{\text{eff}} C (\delta')^2 \right]_0^{h/2} \\
&= \frac{1}{8} L_{\text{eff}} C h^2 + \frac{1}{8} L_{\text{eff}} C h^2
\end{aligned}$$

(26)

### B.3. Comparison and Conclusion

Summing the terms, we obtain the final error bound for the bi-anchor strategy:

$$\mathcal{E}_{\text{BA}}(h) \approx \frac{1}{4} L_{\text{eff}} C h^2. \tag{27}$$

Comparing this to the single-anchor error bound derived in Eq 23, $\mathcal{E}_{\text{SA}}(h) \approx \frac{1}{2} L_{\text{eff}} C h^2$, we observe that:

$$\mathcal{E}_{\text{BA}}(h) \approx \frac{1}{2} \mathcal{E}_{\text{SA}}(h). \tag{28}$$

The bi-anchor mechanism reduces the error bound coefficient by a factor of **2** (and the integrated squared error contribution by a factor of 4 relative to the interval split). By ensuring that the SideNet never predicts across an offset larger than $h/2$, the bi-anchor solver effectively minimizes the accumulation of extrapolation drift. This theoretical reduction validates our empirical findings that BA-Solver can maintain high fidelity even at larger interval sizes.

## C. Error Comparison Between Extrapolation and Interpolation Solvers

In this section, we provide a theoretical comparison of the convergence orders between standard extrapolation solvers and our proposed interpolation-based BA-Solver. We demonstrate that by leveraging high-order quadrature rules within the interpolation framework, BA-Solver achieves a significantly higher algebraic precision than state-of-the-art extrapolation methods.

### C.1. Extrapolation Solvers (Euler & DPM-Solver++)

Extrapolation solvers approximate $\boldsymbol{x}_{t-h}$ by expanding the velocity field around the current time $t$ using Taylor series.

**Euler Solver (1st Order Extrapolation).** The Euler method utilizes a zeroth-order Taylor expansion of the velocity, assuming $\boldsymbol{v}_\tau \approx \boldsymbol{v}_t$ for $\tau \in [t - h, t]$. The local truncation error (LTE) is bounded by the second derivative of the trajectory:

$$\mathcal{E}_{\text{Euler}}(h) = \left\| \int_{t-h}^{t} (\boldsymbol{v}_\tau - \boldsymbol{v}_t) \mathrm{d}\tau \right\| \leq \frac{h^2}{2} \sup_\tau \|\ddot{\boldsymbol{x}}_\tau\| = \mathcal{O}(h^2).$$

(29)

This indicates a global convergence order of 1, which necessitates very small interval sizes $h$ to suppress error.

**DPM-Solver++ (High-Order Extrapolation).** DPM-Solver++ (Lu et al., 2022) improves upon Euler by utilizing historical states to approximate higher-order derivatives of the velocity. A $k$-th order DPM-Solver essentially constructs a $(k - 1)$-th degree polynomial extrapolation. For the commonly used DPM-Solver-3 ($k = 3$), the LTE is:

$$\mathcal{E}_{\text{DPM}}(h) = \mathcal{O}(h^{k+1}) = \mathcal{O}(h^4). \tag{30}$$

While efficient, extrapolation methods inherently suffer from the *"shooting blind"* problem: the error term depends solely on the derivatives at the start point $t$. As $h$ increases, the validity of the Taylor approximation at $t$ degrades rapidly for the distant point $t - h$.

### C.2. Interpolation Solver (BA-Solver)

BA-Solver fundamentally diverges from extrapolation paradigms by formulating the generation step as a numerical integration task. By leveraging the SideNet to accurately predict intermediate velocities, we can employ the Gauss-Lobatto quadrature rule. For a 4-point Gauss-Lobatto rule (comprising 2 endpoints and 2 intermediate nodes), the scheme integrates polynomials of degree up to $2n - 3 = 5$ exactly. Consequently, the theoretical LTE of the numerical scheme is given by:

$$\mathcal{E}_{\text{scheme}}(h) \approx C \cdot h^{2n-1} \cdot \boldsymbol{v}^{(2n-2)}(\xi) = \mathcal{O}(h^7), \tag{31}$$

where $n = 4$ denotes the number of nodes and $v^{(2n-2)}$ represents the higher-order derivative of the velocity field.

Crucially, this scaling behavior highlights a fundamental advantage: while standard extrapolation error depends on the Lipschitz constant $L$ (first-order derivative bound) and scales as $\mathcal{O}(h^2)$, our interpolation scheme benefits from higher-order smoothness, scaling with a significantly more favorable power $\mathcal{O}(h^7)$.

### C.3. Theoretical Comparison and Discussion

Comparing the error bounds, we observe a substantial gap in algebraic precision between the methods:

- **Euler (Extrapolation):** $\mathcal{O}(h^2)$

- **DPM-Solver-3 (Extrapolation):** $\mathcal{O}(h^4)$

- **BA-Solver (Interpolation):** $\mathcal{O}(h^7)$ (Scheme Order)

This analysis elucidates why BA-Solver outperforms extrapolation in the few-step regime (*e.g.*, 5-10 NFEs). While extrapolation solvers are inherently limited by the local derivative information at $t$ ("shooting blind"), BA-Solver exploits the global structure of the interval via quadrature.

**Remark on Error Composition.** We explicitly clarify that the BA-solver's **total error is indeed bounded by** $\mathcal{O}(h^2)$. Our empirical superiority over standard standard solvers comes fundamentally from a highly compressed constant prefactor, driven by three key factors:

1. **Guaranteeing Integration Precision:** By employing a 4-point Gauss-Lobatto quadrature ($\mathcal{O}(h^7)$ scheme error), the numerical discretization error becomes virtually negligible, safely isolating the generation error to the network approximation and systematic drift.

2. **Smaller Effective Constant ($L_{\text{eff}}$):** Standard solvers' error bounds depend on the Lipschitz constant $L$. Our isolated network error behaves empirically as if governed by a much smaller $L_{\text{eff}}$, representing only the uncaptured state drift.

3. **Reducing the Prefactor via Bi-Anchor:** The bi-anchor mechanism strictly limits the maximum prediction offset to $h/2$, mathematically scaling down the integrated bound of the dominant error term.

## D. More Related Work

**Image Generation.** Image generation has advanced rapidly over the past decade, motivated by the pursuit of both tractable likelihoods and high-fidelity synthesis. Early approaches such as Variational Autoencoders (VAEs) (Kingma

*Table 4.* **Hyperparameters for SideNet architecture and training configurations.** Parameters are reported for both ImageNet-$256^2$ and ImageNet-$512^2$ experiments. Common settings are merged for brevity.

| Hyperparameter | Resolution | |
|---|---|---|
| | ImageNet-$256^2$ | ImageNet-$512^2$ |
| *SideNet Architecture* | | |
| Input Channels | 4 | |
| Base Channels | 128 | |
| Time Embed. Dim. | 256 | |
| Number of Layers | 4 | 8 |
| *Training Configuration* | | |
| Integration Rule | Gauss-Legendre | |
| Truncated Exp. $\lambda$ | 50 | |
| Chain Length | 8 | |
| Batch Size | 4096 | |
| Training Iterations | 0.25K | 0.50K |
| Learning Rate | $1 \times 10^{-4}$ | |

& Welling, 2013) optimize the evidence lower bound (ELBO), but are often limited by overly smooth or blurry samples. Subsequent progress was marked by Denoising Diffusion Probabilistic Models (DDPMs) (Ho et al., 2020), which model the generative process as the reversal of a gradual noise-adding Markov chain. While DDPMs significantly improved sample quality, their inference procedure originally required a large number of iterative steps. Denoising Diffusion Implicit Models (DDIMs) (Song et al., 2020a) addressed this limitation by introducing a non-Markovian formulation that enables more efficient, and in certain cases deterministic, sampling. These diffusion-based approaches were later unified under the framework of Score-based Generative Models (SGMs) (Song et al., 2020b), which provide a continuous-time perspective via stochastic differential equations (SDEs).

More recently, flow matching (Lipman et al., 2022) has been proposed as an alternative paradigm for training Continuous Normalizing Flows (CNFs) without explicitly simulating stochastic trajectories. In contrast to diffusion models that rely on SDE-driven stochastic paths, flow matching directly learns a time-dependent velocity field that transports a simple prior distribution (e.g., a Gaussian) to the data distribution through an ordinary differential equation (ODE). By constructing training targets along Optimal Transport (OT) paths, flow matching often yields straighter transport trajectories, which has been shown to facilitate optimization and enable more efficient inference in practice.

**Flow Matching Backbones.** The effectiveness of diffusion and flow matching models is strongly influenced by the neural architectures used to parameterize the underlying time-dependent vector field or score functions. Early diffusion models predominantly relied on U-Net architectures (Ronneberger et al., 2015), whose hierarchical convolutional structure and multiscale inductive biases proved well-suited for pixel-space image generation (Dhariwal &

*Table 5.* **Sampling hyperparameters for different NFEs.**

| Parameter | Number of Function Evaluations (NFE) | | | | | | | |
|---|---|---|---|---|---|---|---|---|
| | 5 | | 7 | | 10 | | 15 | |
| | $256^2$ | $512^2$ | $256^2$ | $512^2$ | $256^2$ | $512^2$ | $256^2$ | $512^2$ |
| CFG Scale | $[0, 0.7]$ | $[0, 0.8]$ | $[0, 0.7]$ | $[0, 0.8]$ | $[0, 0.8]$ | $[0, 0.8]$ | $[0, 0.8]$ | $[0, 0.8]$ |
| CFG Interval | 3.1 | 3.5 | 2.8 | 2.7 | 2.0 | 2.3 | 2.0 | 1.95 |

*Table 6.* **Quantitative comparison on class-conditional ImageNet-$256^2$ and ImageNet-$512^2$.** We compare BA-solver against other training-free solvers under NFE=15, 10, 7 and 5. ↑/↓ indicates higher/lower is better. Best results in each block are marked in **bold**.

| Methods | ImageNet-$256^2$ | | | | | ImageNet-$512^2$ | | | | |
|---|---|---|---|---|---|---|---|---|---|---|
| | IS ↑ | FID ↓ | sFID ↓ | Prec. ↑ | Rec. ↑ | IS ↑ | FID ↓ | sFID ↓ | Prec. ↑ | Rec. ↑ |
| *Number of Function Evaluations (NFE) = 5* | | | | | | | | | | |
| Euler-Solver (Stoer et al., 1980) | 214.59 | 11.58 | 12.01 | 0.66 | 0.48 | 203.20 | 13.12 | 15.20 | 0.71 | 0.44 |
| Heun-Solver (Stoer et al., 1980) | 238.31 | 7.88 | **7.04** | 0.70 | 0.51 | 114.00 | 26.11 | 17.97 | 0.59 | 0.53 |
| Flow-UniPC (Zhao et al., 2023) | 243.64 | 9.43 | 10.85 | 0.70 | 0.42 | 127.42 | 23.08 | 18.23 | 0.62 | 0.48 |
| Flow-DPM (Xie et al., 2024) | 261.90 | 6.18 | 7.04 | 0.73 | 0.52 | 256.66 | 8.20 | **10.67** | 0.76 | 0.47 |
| **BA-solver (Ours)** | **306.22** | **2.84** | 8.09 | **0.76** | **0.63** | **270.26** | **5.18** | 12.45 | **0.78** | **0.60** |
| *Number of Function Evaluations (NFE) = 7* | | | | | | | | | | |
| Euler-Solver (Stoer et al., 1980) | 238.54 | 7.56 | 9.09 | 0.71 | 0.55 | 204.68 | 11.04 | 13.17 | 0.73 | 0.52 |
| Heun-Solver (Stoer et al., 1980) | 220.51 | 6.89 | 7.72 | 0.70 | 0.60 | 214.10 | 9.98 | 11.17 | 0.73 | 0.54 |
| Flow-UniPC (Zhao et al., 2023) | 258.89 | 5.09 | 6.12 | 0.74 | 0.57 | 243.45 | 8.43 | 9.20 | 0.75 | 0.49 |
| Flow-DPM (Xie et al., 2024) | 280.16 | 4.03 | 7.74 | 0.76 | 0.58 | 255.34 | 6.59 | 10.99 | 0.77 | 0.54 |
| **BA-solver (Ours)** | **320.49** | **1.96** | **6.06** | **0.78** | **0.64** | **300.79** | **2.88** | 8.35 | **0.80** | **0.61** |
| *Number of Function Evaluations (NFE) = 10* | | | | | | | | | | |
| Euler-Solver (Stoer et al., 1980) | 290.26 | 3.84 | 6.31 | 0.79 | 0.55 | 269.02 | 5.08 | 9.86 | 0.78 | 0.54 |
| Heun-Solver (Stoer et al., 1980) | 299.93 | 2.96 | 5.61 | 0.81 | 0.56 | 262.29 | 5.64 | 8.83 | 0.78 | 0.54 |
| Flow-UniPC (Zhao et al., 2023) | 316.97 | 3.27 | 6.09 | **0.83** | 0.52 | 290.84 | 5.38 | 7.23 | 0.79 | 0.50 |
| Flow-DPM (Xie et al., 2024) | **318.08** | 2.62 | 6.27 | 0.82 | 0.57 | **304.49** | 3.85 | 8.83 | **0.80** | 0.55 |
| **BA-solver (Ours)** | 311.16 | **1.72** | **4.92** | 0.80 | **0.63** | 286.38 | **2.16** | **6.17** | 0.80 | **0.63** |
| *Number of Function Evaluations (NFE) = 15* | | | | | | | | | | |
| Euler-Solver (Stoer et al., 1980) | 326.91 | 2.56 | 4.76 | 0.82 | 0.57 | 266.41 | 3.61 | 8.57 | 0.80 | 0.57 |
| Heun-Solver (Stoer et al., 1980) | 314.36 | 1.99 | 5.42 | 0.81 | 0.61 | 250.30 | 3.58 | 7.82 | 0.79 | 0.61 |
| Flow-UniPC (Zhao et al., 2023) | 327.72 | 2.59 | 8.37 | 0.82 | 0.58 | 272.55 | 3.34 | 7.99 | 0.80 | 0.57 |
| Flow-DPM (Xie et al., 2024) | **341.85** | 2.32 | 5.49 | **0.83** | 0.58 | **287.26** | 2.79 | 7.79 | **0.80** | 0.58 |
| **BA-solver (Ours)** | 326.67 | **1.65** | **4.43** | 0.81 | **0.62** | 272.14 | **1.83** | **5.04** | 0.80 | **0.63** |

Nichol, 2021). More recently, the success of Vision Transformers (ViTs) (Dosovitskiy, 2020) has motivated a shift toward attention-based architectures (Vaswani et al., 2017) in generative modeling.

Diffusion Transformers (DiT) (Peebles & Xie, 2023) demonstrate that replacing U-Net backbones with standard Transformer architectures can substantially improve scalability and performance, exhibiting scaling behavior analogous to that observed in large language models. Building upon this line of work, SiT (Ma et al., 2024) explores scale-wise interpolation strategies that relate diffusion prediction targets to flow matching objectives. In the context of flow matching, Transformer-based backbones, including DiT and its variants, are increasingly adopted due to their capacity to model global interactions and long-range dependencies, which are critical for learning high-dimensional vector fields in text-to-image generation tasks.

## E. Setting and More Results

**Training Details.** We detail the hyperparameter settings for the SideNet architecture and training configurations in Table 4. SideNet is designed as a lightweight, conditional network that predicts a velocity correction term by processing the concatenated inputs of the noisy state $x_t$ and the base model's velocity $v_t$. The architecture incorporates multi-modal conditioning—integrating timestep $t$, interval size $h$, and class labels $y$—which modulates feature maps via Feature-wise Linear Modulation (FiLM) coupled with parameter-free Channel RMSNorm. To achieve efficiency, the backbone comprises exclusively lightweight depthwise-separable convolutional ResBlocks. The architecture integrates timestep $t$, interval size $h$, and class labels $y$ to modulate features via FiLM and parameter-free Channel RMSNorm. The backbone is built entirely from ultra-lightweight depthwise-separable convolutions.

**Sampling Configuration.** The detailed inference hyperparameters, including Classifier-Free Guidance (CFG) scales and intervals, are presented in Table 5. To ensure a fair

*Table 7.* **Detailed Efficiency Comparison on class-conditional ImageNet-**$256^2$**.** We compare training cost (iterations and trainable parameters) and generation quality (FID) with training-based one- or few-step methods. (M) indicates Million Parameters.

| Method | FID ↓ | Iterations | Training(M) | NFE | Type |
|---|---|---|---|---|---|
| Base Model (REPA+SiT) | 6.92 | 250K | 675 | 8 | pretrained model |
| Base Model (REPA+SiT) | 2.56 | 250K | 675 | 15 | pretrained model |
| Shortcut (Frans et al., 2024) | 10.60 | 78.0K | 675 | **1** | train from scratch |
| MeanFlow (DiT) (Geng et al., 2025a) | 3.43 | 313K | 676 | **1** | train from scratch |
| $\pi$-Flow-NFE-1 (Chen et al., 2025) | 2.85 | 140K | 676 | **1** | Distillation from REPA+SiT |
| alpha-flow (Cheng et al., 2025) | 2.58 | 95.0K | 676 | **1** | train from scratch |
| iCT (Song & Dhariwal, 2023) | 20.30 | 800K | 675 | 2 | train from scratch |
| MeanFlow (DiT) (Geng et al., 2025a) | 2.20 | 313K | 676 | 2 | train from scratch |
| MeanFlow (SiT)[3] (Geng et al., 2025a) | 2.55 | 41.0K | 675 | 2 | fine-tune from pretrained SiT |
| $\pi$-Flow-NFE-2 (Chen et al., 2025) | 1.97 | 24.0K | 676 | 2 | distillation from REPA+SiT |
| alpha-flow (Cheng et al., 2025) | 2.15 | 95.0K | 676 | 2 | train from scratch |
| MeanFlow (SiT)[3] (Geng et al., 2025a) | 5.58 | 41.0K | 675 | 8 | fine-tune from pretrained SiT |
| alpha-flow (Cheng et al., 2025) | 4.53 | 95.0K | 676 | 8 | train from scratch |
| **BA-solver (Ours)** | **1.85** | **0.25K** | **6** | 8 | **lightweight** training |

*Table 8.* Real-World Inference Time Per Batch.

| Operation | Evaluations | Time (s) |
|---|---|---|
| Overall Backbone Time | 1 | 4.1837 |
| Overall SideNet Time | 4 | 0.2554 |
| Integration Overhead | - | 0.0112 |
| **Total Time** | **-** | **4.4502** |

comparison, all baselines adopt identical sampling settings.

**Real-World Inference Time Per Batch** Real wall-clock time is considered the ultimate metric for evaluating practical acceleration. The real-world inference time per batch was profiled using the default generation setting (4-point quadrature) on ImageNet-256 (NVIDIA H800 GPU, batch size 256). This setup corresponds to one backbone evaluation and four SideNet evaluations per integration step. The empirical results demonstrate that the SideNet and integration operations introduce an approximate 6% latency overhead per batch. These findings support the initial premise: reducing the total number of sequential backbone evaluations, while incurring minimal SideNet overhead, translates theoretical NFE (Neural Function Evaluation) reductions into meaningful wall-clock speedups. This provides a significant advantage over standard solvers that require multiple sequential backbone evaluations per step. The detailed breakdown of the time required for each operation is presented in Table 8.

**More Results.** A detailed version of Table 2 is shown in Table 7. We present additional visual samples generated by our BA-solver on class-conditional ImageNet-$256^2$ and ImageNet-$512^2$ using a budget of only 7 NFEs, as shown in Figure 7 and Figure 8. Furthermore, we have updated Table 6 to incorporate quantitative results for 10 NFE.

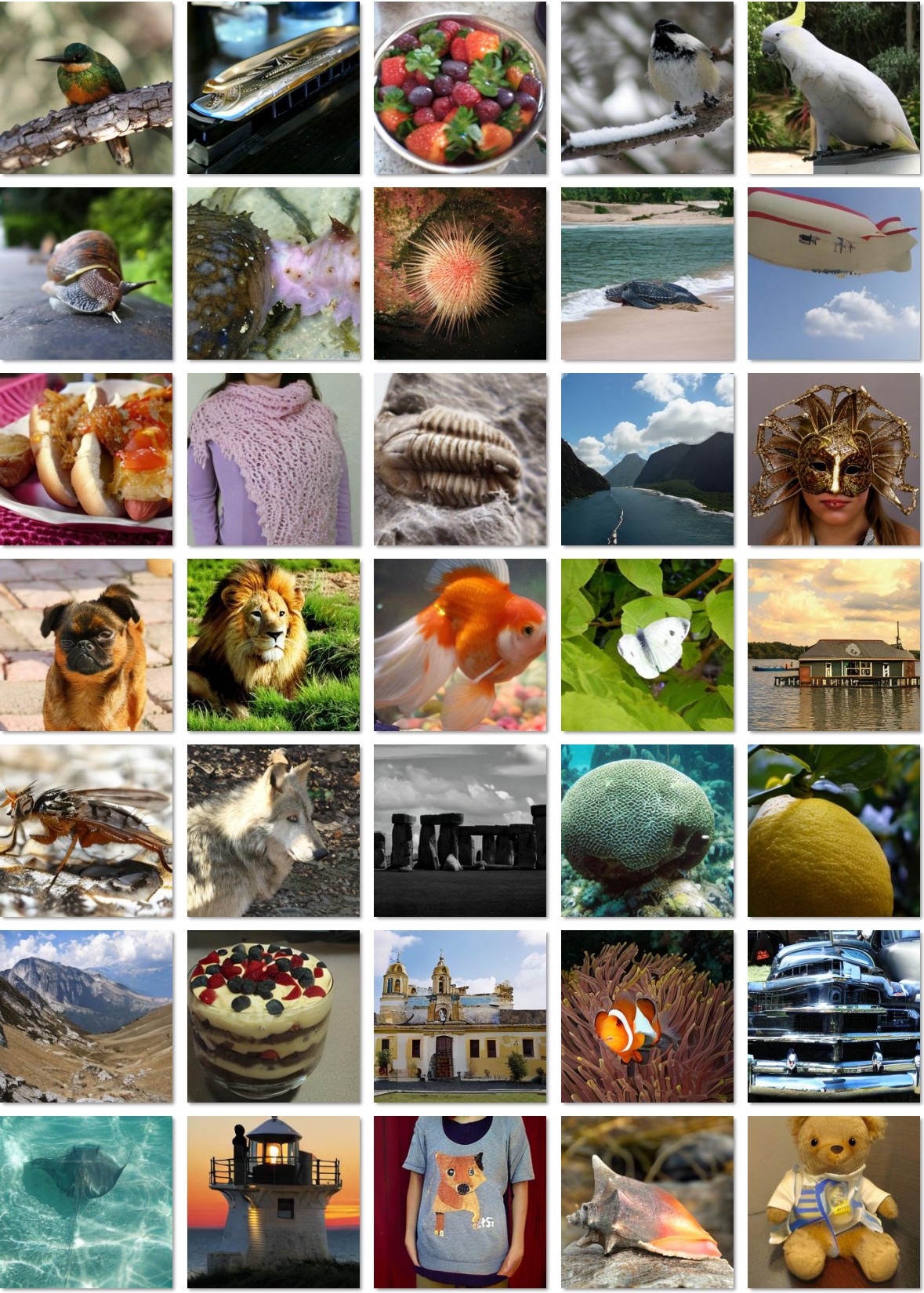

*Figure 7.* More visual samples generated by BA-Solver on class-conditional ImageNet-256$^2$ with only 7 NFEs.

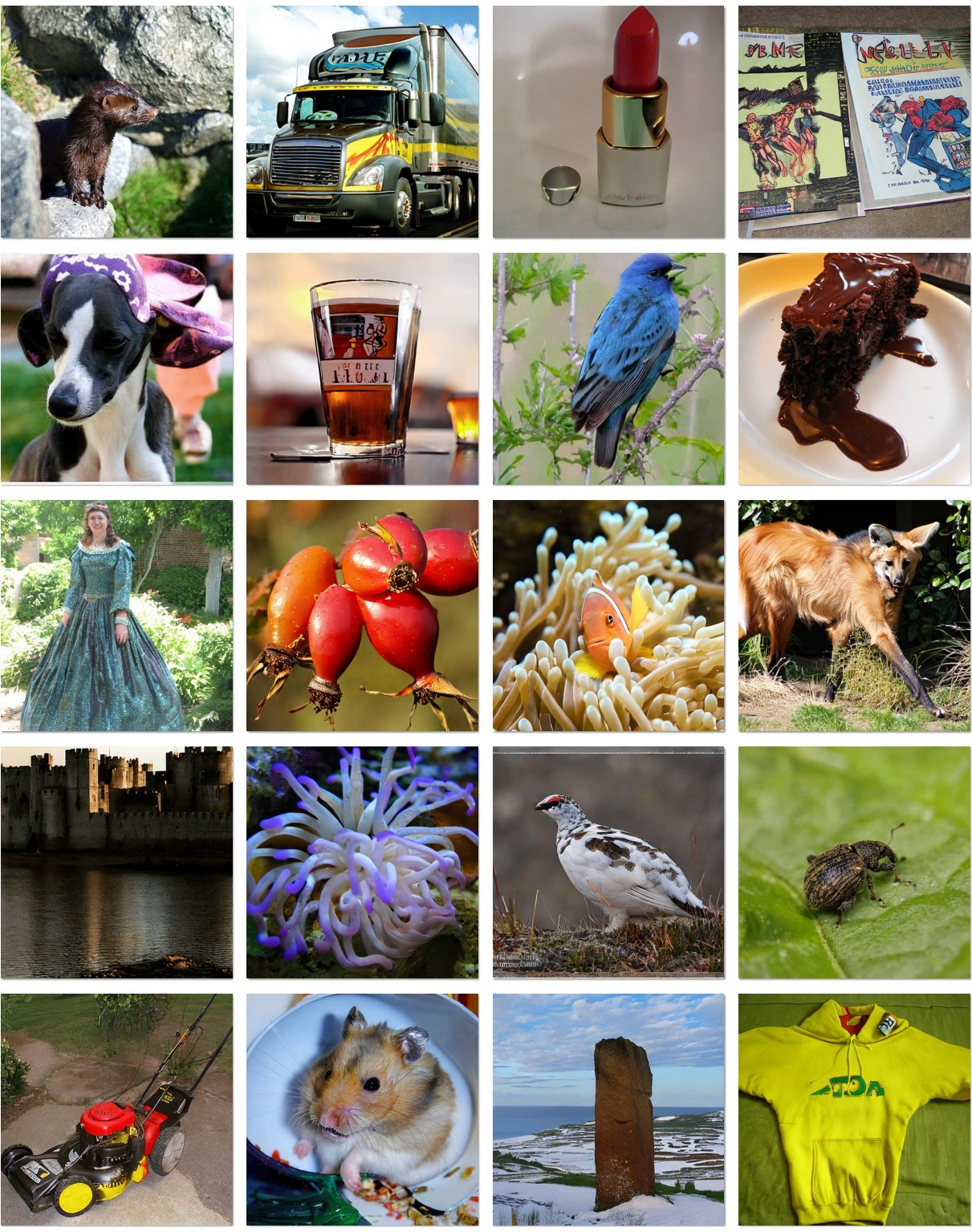

*Figure 8.* More visual samples generated by BA-Solver on class-conditional ImageNet-$512^2$ with only 7 NFEs.

