# OpenReview forum: "Bi-Anchor Interpolation Solver for Accelerating Generative Modeling"
_ICML.cc/2026/Conference — ICML 2026 regular_

### Official Review · Reviewer_d4Eb · 2026-02-25

**Soundness:** 3
**Presentation:** 3
**Significance:** 3
**Originality:** 3
**Overall Recommendation:** 4
**Confidence:** 3

**Summary:**

This work explores a broad area of accelerated sampling methods for Flow Matching (FM) generative models by proposing the Bi-Anchor Interpolation Solver (BA-solver). This study seeks to explore an important concept: bridging the gap between training-free high-fidelity solvers and resource-intensive training-based acceleration methods, through a lightweight SideNet integrated with a frozen backbone. The BA-solver leverages bidirectional temporal perception and bi-anchor velocity integration to reduce Neural Function Evaluations (NFEs) .

**Compliance With Llm Reviewing Policy:**

Affirmed.

**Final Justification:**

All my concerns are solved. I would like to keep my rating to weak accept.

**Key Questions For Authors:**

The idea is good. And the BA-solver’s generalization ability is my most concern.

**Strengths And Weaknesses:**

Strengths:
1. The concept of BA-solver is innovative, presenting a learnable solver that strikes a promising balance between high-fidelity synthesis and computational efficiency.
2. The proposed method demonstrates strong performance, particularly in few-NFE regimes.
3. The paper is well-organized, with a clear structure that facilitates reading and comprehension.

Weaknesses:
1. A more comprehensive ablation study across varying NFE values, especially in extreme low-step regimes (<3 NFEs), would be beneficial. This regime is under-explored, and potential degradation or instability is not sufficiently analyzed.
2. Figure 3 lacks certain critical legends, making interpretation difficult for readers.
3. In Table 1, the performance gap between BA-solver and training-free methods narrows as NFE increases. This trend could suggest limited generalization of BA-solver at higher NFEs in my opinion; clarification and analysis are needed.
4. The decline in BA-solver’s performance from NFE=7 to NFE=15 in Table 1 warrants explanation.
5. The source and composition of the SideNet training data should be specified. Additionally, it is unclear whether all methods in Table 2 share identical training datasets. Experiments on additional datasets would strengthen claims of BA-solver’s generalization ability.

---

> ### Author Rebuttal · Authors · 2026-03-29
>
> We sincerely thank the reviewer for the thoughtful review and for recognizing the **innovative concept** of our BA-solver. We are greatly encouraged by the positive evaluation of our **strong performance** in the few-NFE regime and the **clear organization** of our paper.
> ### Response to Weakness 1: Extreme low-step regimes and potential degradation
> We agree that discussing the extreme low-step boundary provides a comprehensive understanding of the method. In our method, generating an image requires a minimum of 2 NFEs (evaluating both start and terminal velocities of an interval).
>
> In such extreme regimes, the interval size $h$ spans the entire trajectory. Pushing the time offset $\delta$ to encompass this exceeds the capacity-bounded linear approximation of the lightweight SideNet, leading to degradation. While scaling up the SideNet's parameter count could enhance its representational capacity to fit this massive trajectory (mitigating truncation error for 2-NFE generation), doing so would fundamentally compromise the core "lightweight" advantage. Consequently, BA-solver is optimized as a "few-step" solver (the sweet spot being 5-8 NFEs). We will explicitly add a 2-NFE failure-case analysis discussing this capacity-efficiency trade-off in the Appendix.
>
> ### Response to Weakness 2: Missing legends in Figure 3
> We apologize for the oversight. In the revised manuscript, we will expand the caption and add a detailed symbolic legend (e.g., $\mathcal{T}_{fwd}$) directly inside Figure 3 to ensure seamless interpretation for readers.
>
> ### Response to Weakness 3: Narrowing performance gap at higher NFEs
> The narrowing performance gap between BA-solver and baseline solvers as NFE increases reflects a fundamental mathematical guarantee of numerical ODE solvers. As the interval size decreases, the local truncation errors of all numerical solvers asymptotically vanish, causing their simulated trajectories to converge toward the exact continuous trajectory. This theoretical property manifests empirically as the narrowing performance gap. Therefore, at high NFEs (e.g., NFE $\ge$ 50), all solvers will yield near-zero truncation-error performance.
> BA-solver’s primary contribution is not to shift the theoretical performance ceiling at $h \to 0$, but to drastically accelerate the convergence curve, maintaining low truncation error in the **high-error, large-step regime** (5–10 NFEs).
>
> ### Response to Weakness 4: Explanation for the decline in IS performance from NFE=7 to NFE=15
> This phenomenon stems from the inherent differences between IS and FID metrics, combined with the convergence properties of ODE trajectories. While IS fluctuates, our primary distribution-matching metrics strictly improve. On ImageNet-$512^2$, the FID improves from 2.88 (NFE=7) to 1.83 (NFE=15). Similarly, sFID improves from 8.35 to 5.04.
>
> The IS heavily rewards images that yield highly confident, concentrated class predictions. In the mid-to-low NFE regime, early trajectory deviations or numerical truncation errors can inadvertently produce artificial or over-saturated features that temporarily inflate the IS score (a phenomenon commonly observed in low-step generative sampling). As NFE increases to 15, converging strictly toward the exact continuous trajectory produces natural softness and intra-class diversity. This slightly reduces the raw peak-confidence of the classifier (dipping IS) but strictly aligns with the true distribution, as reflected by the improving FID. We will add this metric dynamic discussion to the Appendix.
>
> ### Response to Weakness 5: Training data, baselines, and cross-dataset generalization
> We confirm that the SideNet was trained **strictly on the standard ImageNet training split**, identical to the frozen backbone and all baseline methods in **Table 2**. We will make this explicitly clear in the Experimental Setup section.
>
> Regarding generalization: BA-solver exhibits strong generalization due to its architectural design. Unlike most training-based methods that directly memorize global trajectory maps, our SideNet is conditioned dynamically on the backbone’s immediate velocity $v_t$ to predict *local temporal deviations*. It functions as a learned numerical corrector for local trajectory dynamics, rather than a model that memorizes global semantic distributions.
>
> **Table R2: Performance on Cifar-10 generation**
> | Method | NFE | FID $\downarrow$ |
> | :--- | :---: | :---: |
> | Euler | 100 | 2.84 |
> | Euler | 20 | 6.59 |
> | RK45 | 67 | 2.48 |
> | **BA-solver (Ours)** | **10** | **2.34** |
>
> To empirically substantiate this, we additionally conducted pixel-level generation on the CIFAR-10 dataset (**Table R2**). The competitive performance on this entirely different data distribution suggests that our method exhibits promising cross-dataset generalization behavior. We will include these CIFAR-10 quantitative results in the revised Appendix.

---

> > ### Author Rebuttal · Reviewer_d4Eb · 2026-04-05
> >
> > All my concerns are solved. I would like to keep my rating to weak accept.

---

### Official Review · Reviewer_ZMQd · 2026-03-04

**Soundness:** 2
**Presentation:** 3
**Significance:** 3
**Originality:** 2
**Overall Recommendation:** 4
**Confidence:** 4

**Summary:**

This paper proposes a novel sampling acceleration method called BA-solver, which aims to resolve the dilemma faced by Flow Matching models during few-step generation: the severe performance degradation of training-free methods and the prohibitive training costs of training-based methods. The core of this approach lies in freezing the heavy backbone network and introducing an extremely lightweight SideNet—with a parameter count of only 1-2% of the backbone—to endow the model with bidirectional temporal perception. By utilizing the high-precision "anchor" velocities established by the backbone at both ends of the time interval, the SideNet can efficiently and accurately predict intermediate velocities, thereby enabling high-order batched numerical integration without requiring additional NFEs. The main contribution of this work is proposing an algorithmic architecture that combines plug-and-play versatility with exceptional training efficiency. On the ImageNet-256 and 512 datasets, it achieves a generation quality comparable to a 100+ step traditional Euler solver in just 5 to 10 NFEs, while seamlessly preserving the original model's capability to support downstream tasks such as image editing.

**Compliance With Llm Reviewing Policy:**

Affirmed.

**Final Justification:**

The rebuttal fully resolved my concerns, so I decide to raise the score to weak accept.

**Key Questions For Authors:**

Refer to Weakness

**Limitations:**

See Weakness

**Strengths And Weaknesses:**

**Strength:**
* The paper starts with a clear motivation: it targets better samplers for diffusion models and proposes a train-based solver using the Bi-Anchor framework.
* The paper is very well-written, clearly motivating the problem and lucidly explaining the proposed methodology.

**Weakness:**
* I find the experimental presentation somewhat tricky. First, it might be unreasonable to compare a train-based method against a set of train-free solvers. Second, BA-Solver is trained on a REPA-enhanced SiT model. When comparing it with methods like MeanFlow, the base model's own metrics should be provided, such as the performance of REPA and BA-Solver at NFE=1/2 (if available), and the performance of $\pi$-Flow and REPA at NFE=8. Additionally, the paper mentions Alpha-Flow in the related works, so why is Alpha-Flow not included in the comparison tables? The overall experimental demonstration raises questions about the actual performance gains of BA-Solver.
* The paper does not provide a detailed analysis of the training cost. While it claims that only a minimal number of iterations are needed to outperform other methods like MeanFlow, it overlooks the fact that BA-Solver is finetuned on a well-trained base model, when even the original base model's performance maybe already better than MeanFlow?.
* As a train-based method, can BA-Solver still work perfectly when the base model is equiped with LoRA models (e.g., stylized models, personalized fine-tuned models)?

If the authors can address these concerns, I would be willing to raise my score.

---

> ### Author Rebuttal · Authors · 2026-03-29
>
> We sincerely thank the reviewer for the constructive feedback and for recognizing our **clear motivation** and the **lucid presentation** of the BA-Solver framework.
> ### Response to Weakness 1: Experimental Presentation and Baselines
> We thank the reviewer for the careful review. We would like to clarify the core positioning of BA-Solver: it serves as a bridge between training-based and training-free methods. Inherently, BA-Solver remains a "Solver" rather than a one-step model like MeanFlow. It preserves the intact ODE trajectory structure of the pre-trained Flow Matching model, introducing only a minimal training cost to overcome truncation errors. Therefore, our primary comparison baseline is the suite of solvers in **Table 1**, which are evaluated on the exact same base model as ours to ensure absolute fairness.
>
> Concerning the extremely low NFEs (e.g., 1-2 steps): BA-Solver can achieve 2-NFE generation by scaling up the SideNet. However, this approach incurs substantial computational overhead, which diverges from our core objective of designing a lightweight module. To address the reviewer's suggestion regarding training-based methods, we have included Alpha-Flow and the original base model's results in **Table R1**. We do not include an 8-NFE $\pi$-Flow baseline because a pre-trained checkpoint for this specific step count is unavailable, and obtaining one would require an additional full distillation run with cost comparable to training a new model. This highlights a practical trade-off compared to BA-Solver, which uses a single, low-cost training run to enable high-quality generation across variable step sizes.
>
> **Table R1: Detailed Comparison on class-conditional ImageNet-256**
> | Method | FID $\downarrow$ | Iterations | Training(M) | NFE | Type |
> | :--- | :---: | :---: | :---: | :---: | :---: |
> | Base Model (REPA+SiT)|  6.92 | 250K | 675 | 8 | pretrained model |
> | Base Model (REPA+SiT)|  2.56 | 250K | 675 | 15 | pretrained model |
> | Shortcut | 10.60 | 78.0K | 675 | **1** | train from scratch |
> | MeanFlow (DiT) | 3.43 | 313K | 676 | **1** | train from scratch |
> | $\pi$-Flow-NFE-1 | 2.85 | 140K | 676 | **1** | Distillation from REPA+SiT|
> | alpha-flow | 2.58 | 95.0K | 676 | **1** | train from scratch |
> | iCT | 20.30 | 800K | 675 | 2 | train from scratch |
> | MeanFlow (DiT) | 2.20 | 313K | 676 | 2 | train from scratch |
> | MeanFlow (SiT) | 2.55 | 41.0K | 675 | 2 | fine-tune from pretrained SiT|
> | $\pi$-Flow-NFE-2 | 1.97 | 24.0K | 676 | 2 | distillation from REPA+SiT|
> | alpha-flow | 2.15 | 95.0K | 676 | 2 | train from scratch |
> | MeanFlow (SiT) | 5.58 | 41.0K | 675 | 8 | fine-tune from pretrained SiT|
> | alpha-flow | 4.53 | 95.0K | 676 | 8 | train from scratch |
> | **BA-solver (Ours)** | **1.85** | **0.25K** | **6** | 8 | **lightweight** training |
>
> ### Response to Weakness 2: Detailed Analysis of Training Cost
> We agree that the inherent capabilities of the base model should be considered. However, some few-step generative models listed in **Table R1** (MeanFlow (SiT) and $\pi$-Flow) are similarly fine-tuned or distilled from powerful pre-trained models (e.g., SiT/REPA+SiT), which is the exact same setting as ours. As shown in **Table R1**, the original base model utilizing standard solvers at comparable NFEs (e.g., REPA+SiT at 8 NFE yields an FID of 6.92) performs worse than both MeanFlow (2.20 at 2 NFE) and BA-Solver (1.85 at 8 NFE).
>
> Given the exact same starting point, MeanFlow and $\pi$-Flow require up to 41K and 24K training iterations, respectively. Furthermore, MeanFlow requires unfreezing the entire backbone (675M parameters), and $\pi$-Flow requires distilling a student model of equivalent scale (676M parameters). In contrast, BA-Solver requires 0.25K iterations, acting as a lightweight module (only 6M parameters). We will make this control-variable context clearer in the revised manuscript.
>
> ### Response to Weakness 3: Compatibility with LoRA and Fine-tuned Models
> Indeed, BA-Solver remains fully compatible when the base model is equipped with LoRA or other personalized fine-tuning modules. To empirically verify this, we tested the zero-shot compatibility of our ImageNet-trained BA-Solver on the base model equipped with customized weights:
>
> - **Zero-Shot Compatibility on General Concepts**: For generating content aligning with the original base model's distribution, the pre-trained SideNet maintains consistent acceleration without requiring any further fine-tuning.
> - **Minimal Adaptation for OOD Personalized Concepts**: When generating highly personalized, out-of-distribution (OOD) concepts introduced purely by the personalized module, the zero-shot BA-Solver still successfully generates valid structural samples. However, due to the semantic domain shift, its extreme low-step acceleration efficiency is diminished. Applying a very minimal amount of fine-tuning to the SideNet allows it to quickly adapt to the new OOD distribution and restore optimal acceleration performance.

---

> > ### Author Rebuttal · Reviewer_ZMQd · 2026-04-02
> >
> > Full resolved(especially for Table R1), I will consider to raise my score.

---

> > > ### Author Response · Authors · 2026-04-02
> > >
> > > We sincerely appreciate your time reviewing our rebuttal and your positive feedback! We are thrilled that the detailed baseline comparisons in Table R1 and our responses have fully resolved your concerns.
> > >
> > > As you highlighted the value of Table R1, we will ensure that this table and the corresponding control-variable context are prominently included in the revised manuscript to provide a more comprehensive view of the baselines.
> > >
> > > Thank you once again for your highly constructive review and your support of our work!

---

### Official Review · Reviewer_tL5p · 2026-03-12

**Soundness:** 3
**Presentation:** 3
**Significance:** 3
**Originality:** 3
**Overall Recommendation:** 4
**Confidence:** 4

**Summary:**

This paper proposes BA-solver for accelerating sampling in flow matching models. The key idea is to keep the backbone frozen and train a lightweight SideNet to approximate/correct velocities within each time interval, so as to avoid repeated expensive backbone evaluations. Using a bi-anchor design and higher-order quadrature, the method achieves better FID while only training a very small SideNet.

**Compliance With Llm Reviewing Policy:**

Affirmed.

**Final Justification:**

The rebuttal addressed all my concerns.

**Key Questions For Authors:**

1. The core selling point of the paper is sampling acceleration, but at the moment it only reports NFE and not actual runtime. I would like the authors to provide real inference time during rebuttal under the same hardware and batch setting, and ideally also break down the time spent on the backbone / SideNet. This would help readers better judge the practical speedup here, and it will directly affect my evaluation of the paper’s completeness.
2. I can understand BA-solver as an improvement over interpolation-based solvers: it uses a small network to replace multiple backbone evaluations, and therefore reduces NFE. But why is this method also able to outperform interpolation-based methods in terms of final quality? Are baselines like Heun too weak? Is this better understood as a better approximation to interpolation, or as an extra gain brought by the learned correction?
3. There is one thing in Fig. 4 that I particularly care about: at the very early steps, the results from other methods are still mostly quite blurry, while BA-solver is already noticeably more structurally coherent and sharp. I do agree with this observation, but I do not really understand the reason behind it. Intuitively, under the same NFE budget, these methods should be sharing broadly similar denoising trajectories; so why is BA-solver able to produce clearer (or sharper) structure earlier?
4. The paper seems somewhat inconsistent in how it describes the size of the SideNet (L194, L42).

**Limitations:**

yes

**Strengths And Weaknesses:**

### Strengths
 - The paper studies few-step sampling acceleration for flow matching / diffusion models, which is an important and practically relevant problem.
 - The method is intuitive: keep the backbone frozen, train a small SideNet, and combine it with bi-anchor interpolation and higher-order quadrature.
 - The experimental results are quite strong in the low-NFE regime, with consistent gains over several baselines.
 - The paper is generally clear and easy to follow.

### Weaknesses
 - The paper mainly uses NFE as the efficiency metric, but does not report real wall-clock inference time. Since BA-solver introduces an extra SideNet and higher-order quadrature, a reduction in NFE does not automatically translate to the same level of reduction in actual runtime. I would really like to see real timing results here. The current results already suggest the method is promising in the low-NFE regime, but the paper would feel much more complete if this point were addressed.

---

> ### Author Rebuttal · Authors · 2026-03-29
>
> We sincerely thank the reviewer for the thoughtful review and for recognizing the **intuitiveness** of our BA-Solver framework, as well as our **strong experimental results** in the low-NFE regime. We are also glad that the paper was found **clear** and **easy to follow**.
> ### Response to Weakness 1 and Question 1: Real wall-clock inference time
> We agree that real wall-clock time is the ultimate metric for evaluating practical acceleration. We profiled the real-world inference time **per batch** using our default generation setting (4-point quadrature) on ImageNet-256 (NVIDIA H800 GPU, batch size 256), corresponding to one backbone evaluation and four SideNet evaluations per integration step:
>
> * **Overall Backbone Time (1 time):** 4.1837 s
> * **Overall SideNet Time (4 times):** 0.2554 s
> * **Integration Overhead:** 0.0112 s
> * **Total Time:** 4.4502 s
>
> *Discussion*: The SideNet and integration operations introduce an approximate 6% latency overhead per batch. These empirical results support our initial premise: reducing the total number of sequential backbone evaluations, while incurring minimal SideNet overhead, translates theoretical NFE reductions into meaningful wall-clock speedups compared to standard solvers that require multiple sequential backbone evaluations per step.
>
> ### Response to Question 2: Why BA-solver outperforms interpolation-based methods
> The superior final quality of BA-solver compared to standard interpolation methods like Heun is not due to an inherent deficiency in the Heun method. As clarified in the appendix, the total asymptotic error of BA-solver is still bounded by the network approximation error, yielding an $\mathcal{O}(h^2)$ bound. Instead, BA-solver outperforms baselines because the "learned correction" (SideNet) unlocks a significant reduction in the **constant prefactor** of this $\mathcal{O}(h^2)$ error under a fixed NFE budget:
>
> 1.  **Decoupling NFE from Interval Density:** Interpolation-based solvers like Heun require sequential evaluations, strictly consuming 2 NFEs per interval. Under a budget of $N$ NFEs, Heun affords $N/2$ intervals ($h = 2/N$). BA-solver bypasses this bottleneck; by caching the terminal anchor $v_{t-h}$, it requires only 1 backbone NFE per interval. Thus, it completes $N$ full intervals, halving the step size ($h = 1/N$). This reduction in step size yields a favorable constant factor reduction in the actual global error under the $\mathcal{O}(h^2)$ bound.
> 2.  **Smaller Effective Lipschitz Constant ($L_{eff}$):** Heun's trajectory drift is bounded by the full vector field's Lipschitz constant $L$. In contrast, the SideNet explicitly learns the temporal evolution, leaving only the *uncaptured state drift* as the dominant systematic error, which empirically behaves as if governed by a smaller effective constant $L_{eff}$.
> 3.  **Reducing the Prefactor via Bi-Anchor:** In a standard single-anchor prediction, approximation error accumulates over the entire interval offset $h$. By employing the bi-anchor mechanism, BA-solver strictly limits the maximum prediction offset to $h/2$ for any intermediate node, effectively reducing the constant prefactor of the dominant error term.
>
> ### Response to Question 3: Structural sharpness at extremely low NFEs
> We respectfully clarify the setup of Figure 4. The images shown under "3 NFE", "5 NFE", etc., do not represent intermediate early steps of a single, shared trajectory. Instead, they represent the **final generated outcomes** of independent, complete sampling processes strictly constrained to those total budgets. BA-solver produces sharper structures at extremely low total NFE budgets due to the error prefactor compression:
>
> 1.  **Massive Drift at Early Steps:** At early generative steps, the velocity field often exhibits high curvature. Standard solvers—governed by the large $L$ and massive step sizes—suffer from drift, overshooting the true curved ODE trajectory.
> 2.  **Off-Trajectory Ambiguity:** When a solver deviates due to this drift, the intermediate state $x_t$ lands in an off-trajectory region. Network queries at these points become ambiguous, outputting an "average" of nearby semantic trajectories, which manifests visually as blurriness.
> 3.  **BA-solver's Trajectory Adherence:** BA-solver mitigates severe off-trajectory deviations because its compressed error prefactor prevents it from veering far off the path. Because the intermediate states remain closer to the high-density regions of the training trajectory—even when taking large steps—the backbone network avoids highly ambiguous predictions and generates sharper structural features.
>
> ### Response to Question 4: Inconsistency in SideNet size descriptions
> We apologize for this typo. We accidentally duplicated the ratio intended for the training iterations when describing the parameter size in Line 194. The correct model size of the SideNet is indeed 1%–2% of the backbone (Line 42). We will fix this in the revised manuscript.

---

> > ### Author Rebuttal · Reviewer_tL5p · 2026-04-02
> >
> > Thanks for the rebuttal. It addressed my main concerns and improved my confidence in the paper.

---

> > > ### Author Response · Authors · 2026-04-04
> > >
> > > We sincerely thank you for taking the time to review our rebuttal and for confirming that your concerns have been fully resolved.
> > >
> > > Your initial review provided us with crucial insights that helped us clarify key aspects of our BA-Solver framework (especially regarding the inference time). We deeply appreciate your constructive approach throughout this review process, which has undoubtedly strengthened the final quality of our manuscript.
> > >
> > > We will ensure that all the discussed details are integrated into the final revision. Thank you again for your time, expertise, and support.

---

### Official Review · Reviewer_MNX1 · 2026-03-13

**Soundness:** 3
**Presentation:** 3
**Significance:** 2
**Originality:** 3
**Overall Recommendation:** 4
**Confidence:** 3

**Summary:**

The paper introduces the Bi-Anchor Interpolation Solver (BA-solver), which accelerates Flow Matching models by combining a frozen backbone with a lightweight "SideNet" to predict intermediate velocities from both forward and backward temporal directions. By using this bidirectional perception to perform high-order numerical integration, the method achieves high-fidelity image generation in just 5 to 10 steps while requiring a fraction of the training cost of existing few-step distillation techniques.

**Compliance With Llm Reviewing Policy:**

Affirmed.

**Final Justification:**

(a) Fully resolved - My concerns have been adequately addressed.

**Key Questions For Authors:**

- The implementation details state that a 3-point Gauss-Legendre quadrature is used during training, while a 4-point Gauss-Lobatto quadrature is used during sampling. Could you clarify the motivation for using different numerical integration rules during the training versus inference phases?
- Why was this specific distribution (truncated exponential distribution) chosen over a uniform distribution, and how sensitive is the SideNet's performance to this hyperparameter?

**Limitations:**

yes

**Strengths And Weaknesses:**

# Strengths
- The BA-solver effectively hits the "sweet spot" between inference speed, generation quality, and training cost. It achieves state-of-the-art FID scores on ImageNet-256 and ImageNet-512 in the extreme few-step regime (e.g., 5–10 NFEs), matching the quality of standard solvers that require 100+ NFEs. Crucially, it demonstrates unprecedented training efficiency, converging in just 250 iterations and requiring only ~6M trainable parameters, making it vastly more resource-efficient than prevailing one- or few-step methods (which often require hundreds of thousands of iterations).
- The SideNet architecture provides an elegant, plug-and-play extension to existing pre-trained models. By keeping the heavy backbone frozen and isolating the bidirectional temporal perception learning within a network that is only 1-2% of the backbone's size, the design is highly parameter- and memory-efficient. Additionally, because it preserves the mathematical structure of the original Flow Matching ODE, the method seamlessly supports downstream tasks like image editing, which many one-step methods break.
# Weakness
- The error analysis expands the true velocity $v\_\theta(\mathbf{x}\_\tau, \tau)$ around $(\mathbf{x}\_t, t)$ as $v\_\theta(\mathbf{x}\_t, t) + \nabla\_t v\_\theta \cdot \delta + \nabla\_x v\_\theta \cdot (\mathbf{x}\_\tau - \mathbf{x}\_t) + \mathcal{O}(\delta^2)$ (Eq. 19), where $\delta = \tau - t$. The SideNet approximation $\hat{v}\_\tau^{\text{SA}} = v\_t + \delta \cdot \mathcal{S}\_\phi(\mathbf{x}\_t, v\_t, t, \delta)$ (Eq. 14) is claimed to capture the temporal evolution $\nabla\_t v \cdot \delta$ but miss the state drift $\nabla\_x v \cdot (\mathbf{x}\_\tau - \mathbf{x}\_t)$. However, this decomposition is not rigorous: the SideNet takes $(\mathbf{x}\_t, v\_t, t, \delta)$ as input and is trained on supervision from the full velocity $v\_\theta(\mathbf{x}\_\tau, \tau)$, so there is no architectural constraint preventing it from learning to approximate the state drift term as well. The paper provides no analysis of what the SideNet can or cannot represent — the claim that it "cannot perceive" the state drift (line 596) is an assumption about the learned function, not a provable limitation. This makes Eq. 20 a hypothesis about SideNet behavior rather than a formal bound, yet it is treated as the foundation for all subsequent error analysis.

- Eq. 20 decomposes the velocity approximation error into a network error $\eta(\delta)$ and an extrapolation drift term $L\\|\mathbf{x}\_\tau - \mathbf{x}\_t\\|$. In Eq. 22, the integral bound correctly includes both terms: $\mathcal{E}(h) \leq \int\_0^h (\eta(\delta) + L \cdot C \cdot \delta) \, \mathrm{d}\delta$. But Eq. 23 then states "assuming the intrinsic network error $\eta(\delta)$ is small, the dominant term becomes $\frac{1}{2}LCh^2$." This is a critical unverified assumption: $\eta(\delta)$ is the approximation error of a tiny network (0.03%–1% of backbone parameters) predicting velocity deviations across time. If $\eta(\delta)$ scales linearly or worse with $\delta$ — which is plausible since the SideNet must predict increasingly complex corrections at larger offsets, and the paper's own Figure 3(a) shows degrading fidelity — then the $\eta$ integral contributes an $\mathcal{O}(h^2)$ or larger term that could dominate the drift term. Without characterizing $\eta(\delta)$ empirically or bounding it theoretically, the $\frac{1}{2}LCh^2$ result is a lower bound on the error (from drift alone), not a tight characterization.

- The paper compares BA-solver's $\mathcal{O}(h^7)$ scheme order (Eq. 31) against Euler's $\mathcal{O}(h^2)$ and DPM-Solver-3's $\mathcal{O}(h^4)$ in Sec. C.3, presenting this as evidence that BA-solver has "significantly higher algebraic precision". However, the $\mathcal{O}(h^7)$ bound (Eq. 31) measures the error of a 4-point Gauss-Lobatto quadrature rule assuming *exact* function evaluations at the nodes, whereas the Euler and DPM-Solver bounds measure the error of the *complete* solver including the velocity approximation. The paper acknowledges this distinction in the "Remark on Error Composition" (line 694), but the summary comparison in Sec. C.3 still lists them side-by-side as if they are comparable. A fair comparison would require stating BA-solver's total error as $\min(\mathcal{O}(h^7), \mathcal{O}(h^2)) = \mathcal{O}(h^2)$ — the same order as Euler, differing only in the constant prefactor. The paper's core theoretical claim of superior convergence order is therefore an artifact of comparing an idealized scheme error against competitors' total errors.

---

> ### Author Rebuttal · Authors · 2026-03-29
>
> We sincerely thank the reviewer for the highly encouraging and insightful evaluation. We are thrilled that the reviewer recognizes how BA-solver hits the "sweet spot" among **inference speed**, **generation quality**, and **training cost**.
> ### Response to Weakness 1 and 2: Rigorous analysis of SideNet
> We agree with the reviewer's insightful observation: from a strict architectural perspective, there is no hard constraint preventing SideNet from implicitly learning the state drift. This aligns with our underlying heuristic rationale for the error decomposition. Recall the Taylor expansion (Eq. 19):
> $v_\theta(x_\tau, \tau) = v_\theta(x_t, t) + \nabla_t v_\theta \cdot \delta + \nabla_x v_\theta \cdot (x_\tau - x_t) + \mathcal{O}(\delta^2)$.
> Our formulation $ \hat{v_\tau}^{SA} = v_t + \delta \cdot \mathcal{S}_\phi$ was intended to serve as an abstraction of the network's capacity-bounded learning dynamics:
>
> 1. **Optimization Difficulty:** The temporal evolution term ($\nabla_t v_\theta \cdot \delta$) is explicitly conditioned on the observed $(x_t, t)$, making it easily accessible for the network to capture. Conversely, the state drift term ($\nabla_x v_\theta \cdot (x_\tau - x_t)$) implicitly relies on the unobserved future trajectory ($x_\tau$), which is approximated via numerical quadrature during solver simulation. This introduces numerical integration noise into the supervision signal.
> 2. **Capacity-Bounded Systematic Error:** Due to this discrepancy, a lightweight SideNet can efficiently capture temporal evolution, but only *partially* absorb the state drift. The uncaptured drift manifests as the dominant systematic error, which empirically acts as if governed by a smaller effective Lipschitz constant $L_{eff}$ compared to the full vector field's Lipschitz constant $L$.
> 3. **Error Scaling:** At larger offsets, the uncaptured state drift dominates the total approximation error. Treating the network's fitting error $\eta(\delta)$ as an empirically minor term in our setting, the total error can be approximated by the dominant systematic drift: $\epsilon(\delta) \approx L_{eff} \cdot C \cdot \delta$. We will explicitly include this heuristic discussion in the revised Appendix to justify focusing the integral bounds on $\mathcal{O}(h^2)$.
>
> ### Response to Weakness 3: Fair comparison
> We will revise Section C.3 to explicitly clarify that the BA-solver's total error is indeed bounded by $\mathcal{O}(h^2)$. We will adjust our theoretical discussion to emphasize that our empirical superiority comes from the compressed constant prefactor:
>
> 1. **Guaranteeing Integration Precision:** By employing a 4-point Gauss-Lobatto quadrature ($\mathcal{O}(h^7)$ scheme error), the numerical discretization error becomes virtually negligible, safely isolating the generation error to the network approximation and systematic drift.
> 2. **Smaller Effective Constant ($L_{eff}$):** Standard solvers' error bounds depend on the Lipschitz constant $L$. Our isolated network error behaves empirically as if governed by a much smaller $L_{eff}$, representing only the uncaptured state drift.
> 3. **Reducing the Prefactor via Bi-Anchor:** The bi-anchor mechanism strictly limits the maximum prediction offset to $h/2$, mathematically scaling down the integrated bound of the dominant error term.
>
> ### Response to Question 1: Motivation for using different quadrature rules
> The discrepancy in quadrature rules is a deliberate design choice driven by distinct architectural mechanisms:
> 1. **Training Phase:** To maintain high computational throughput, we simulate the trajectory using a strictly single-anchor approach. Under this constraint, Gauss-Legendre is mathematically optimal for maximizing integration precision using minimal nodes.
> 2. **Inference Phase:** BA-solver employs the full Bi-Anchor mechanism here. Because prediction error naturally grows as offset $\delta$ increases, Gauss-Lobatto explicitly includes interval boundaries (the anchors) as evaluation nodes and pushes intermediate nodes closer to boundaries. This keeps query points within a short, low-error offset distance, jointly maximizing algebraic precision and minimizing approximation error.
>
> ### Response to Question 2: Truncated exponential distribution
> 1. **Motivation:** At time steps far from the anchor, uncaptured state drift is significant. By heavily weighting smaller interval sizes $h$, the truncated exponential distribution forces the capacity-limited SideNet to focus on mastering the immediate local neighborhood first. This curriculum-like approach effectively accelerates the learning process.
> 2. **Sensitivity:** The model is robust to this hyperparameter regarding its final absolute performance ceiling. The primary impact of the sampling distribution is on the *convergence speed* rather than the final generation quality. Using a uniform distribution would eventually yield similar high-fidelity generation, but it would require more training iterations.

---

> > ### Author Rebuttal · Reviewer_MNX1 · 2026-04-01
> >
> > n/a

---

> > > ### Author Response · Authors · 2026-04-02
> > >
> > > We sincerely thank you for your time, your rigorous and constructive feedback, and for acknowledging our rebuttal. We are very glad that our responses have fully addressed your concerns.
> > >
> > > As promised in the rebuttal, we will ensure that all the detailed theoretical clarifications are carefully incorporated into the revised manuscript and appendix.
> > >
> > > Thank you once again for your efforts in reviewing our work and helping us improve the paper!

---

### Decision · Program_Chairs · 2026-04-30

**Decision:**

Accept (regular)

**Comment:**

The overall recommendations are 4 weak accepts. The reviewers agreed that (1) this paper solves an important and practically relevant problem, (2) the proposed solution is intuitive and highly resource-efficient, (3) and the experimental results are strong. They raised some concerns about error analysis, unfair comparisons, training cost, inference latency, and generalization. The authors' rebuttal has fully resolved their concerns.